# MV2Cyl: Reconstructing 3D Extrusion Cylinders from Multi-View Images

**Eunji Hong** [1], **Minh Hieu Nguyen**[1], **Mikaela Angelina Uy**[2,3], **Minhyuk Sung**[1]

[1]Korea Advanced Institute of Science and Technology, [2]Stanford University, [3]NVIDIA

{eunji.hong, hieuristics, mhsung}@kaist.ac.kr
mikaelaangel@nvidia.com

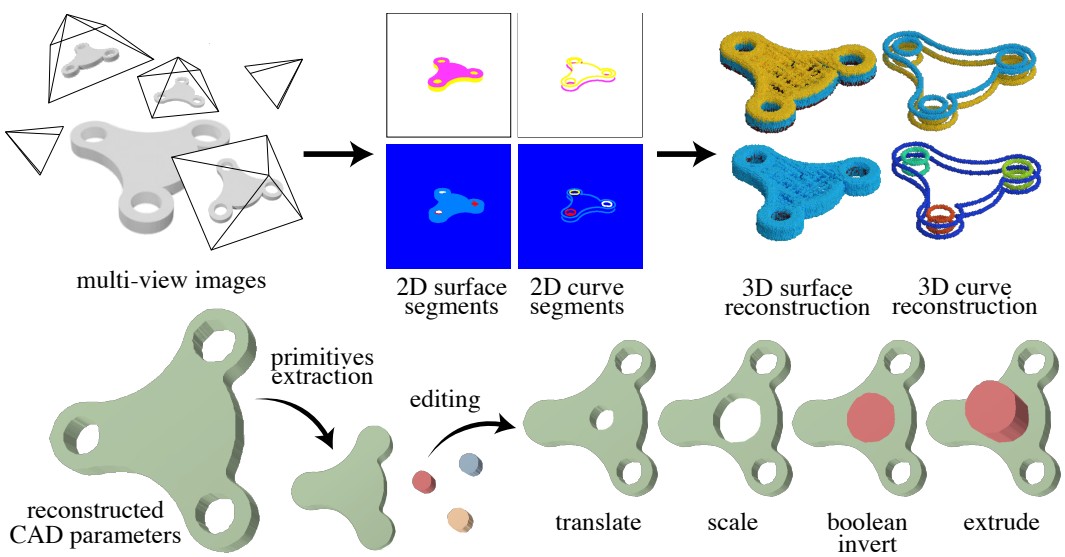

multi-view images    2D surface segments    2D curve segments    3D surface reconstruction    3D curve reconstruction

reconstructed CAD parameters    primitives extraction    editing    translate    scale    boolean invert    extrude

## Abstract

We present MV2Cyl, a novel method for reconstructing 3D from 2D multi-view images, not merely as a field or raw geometry but as a sketch-extrude CAD model. Extracting extrusion cylinders from raw 3D geometry has been extensively researched in computer vision, while the processing of 3D data through neural networks has remained a bottleneck. Since 3D scans are generally accompanied by multi-view images, leveraging 2D convolutional neural networks allows these images to be exploited as a rich source for extracting extrusion cylinder information. However, we observe that extracting only the surface information of the extrudes and utilizing it results in suboptimal outcomes due to the challenges in the occlusion and surface segmentation. By synergizing with the extracted base curve information, we achieve the optimal reconstruction result with the best accuracy in 2D sketch and extrude parameter estimation. Our experiments, comparing our method with previous work that takes a raw 3D point cloud as input, demonstrate the effectiveness of our approach by taking advantage of multi-view images. Our project page can be found at `https://mv2cyl.github.io`.

## 1 Introduction

Most human-made objects in our daily lives are created using computer-aided design (CAD). Reconstructing the structural representation from raw geometry, such as 3D scans, is essential to enable

38th Conference on Neural Information Processing Systems (NeurIPS 2024).

the fabrication of 3D shapes and manipulate them for diverse downstream applications. Sketch-Extrude CAD model is particularly notable not only as the most common but also as a versatile CAD representation, which enables the expression of diverse shapes with 2D sketches and extruding heights.

The reconstruction of a set of extrusion cylinders from raw 3D geometry has been extensively explored in previous studies [58, 65, 49, 30], yet the suboptimal quality of the results remains a challenge. Unsupervised methods [49, 30] have demonstrated notable capabilities in parsing raw geometry but often struggle to precisely segment the shape and part-wise fit the output extrusions to the given shape. Supervised methods using a language-based architecture, such as those described in [65], often face challenges in producing valid outputs. A notable exception is Point2Cyl [58], which is a supervised method proposing to first segment the given raw point cloud into each extrusion region and estimate the 2D sketch and the extrusion parameters using the predicted segments and surface normal information. While it has shown successful results, the bottleneck remains in the limited performance of the 3D backbone network for segmentation.

To address the limitation, we focus on the fact that 3D scans are commonly accompanied by *multi-view images*, which are now even solely used to reconstruct a 3D shape without depth or other information thanks to the recent advancements in neural rendering techniques [40, 61, 13, 62, 5]. 2D multi-view images are a rich resource for extracting 3D extrusion information. Also, 2D convolutional neural networks have been utilized in various 3D tasks [34, 63], owing to their superior performance compared to 3D processing networks. In light of these observations, we introduce a novel framework called MV2Cyl, which reconstructs a set of extrusion cylinders from multi-view images of an object without relying on raw geometry as input.

A straightforward approach to exploiting multi-views in extrusion reconstruction is to first segment 2D regions of the extrusion in each image and reconstruct the 3D shape using the extrusion labels. However, achieving precise 3D surface reconstruction solely from the multi-view images poses challenges due to the ambiguity between the object's intrinsic color (albedo) and the effects of lighting (shading) or shadows within images (Fig. 5). In contrast, we observe that reconstructing only the 2D sketches of the two ends of the extrusion cylinders—the start and end planes of extrusion—provides much more accurate results while avoiding the ambiguity issue and the consequential error accumulation. However, this approach also results in failure cases when one of the bases is not properly detected in 2D image due to the sparse viewpoints. Therefore, we propose a framework that synergizes the reconstructions of labeled surfaces and labeled curves of the extrusion bases so that parameters such as extrusion center and height can be better calculated from the surface information, while the 2D sketches can be precisely recovered from the curve information. In our experiments, we demonstrate the superior performance of our method MV2Cyl on two sketch-extrude datasets: Fusion360 [64] and DeepCAD [65].

## 2   Related Work

**3D Primitive Fitting**. Primitive fitting has been extensively investigated in computer vision. One line of work involves decomposing into surface/volumetric primitives. Classical approaches include detecting simple primitives such as planes [6, 11, 42] through RANSAC [51, 31], region growing [45], or Hough voting [3, 8]. The emergence of neural networks gave rise to data-driven methods [29, 56, 28] learning frameworks that fit simple primitives such as planes, cylinders, cones, and spheres. Primitive fitting has also been used to approach the task of shape abstraction [75, 57] that fit cuboids given an input shape or image. However, these works all assume a fixed predefined set of primitives that may not be able to cover complex real-world objects. The focus of our work is to recover a more general primitive, *i.e.*, extrusion cylinders, that are defined by any arbitrary closed loop and hence can explain more complex objects.

Another line of work tackles fitting 3D parametric curves, where a common strategy is to first detect edges [70] and corners [35, 70, 39] from an input point cloud, and then fit or predict parametric curves from the initial proposals. A recent work, NEF [68], takes multi-view images as input and introduces a self-supervised approach to the parametric curve fitting problem using neural fields, thus allowing them to optimize without needing clean point cloud data. Despite having multi-view images as input, NEF only tackles recovering parametric curves and not extrusion cylinders, which is a basic building block for the sketch-extrude reverse engineering task and is the focus of our work.

**CAD Reconstruction for Reverse Engineering**. CAD reverse engineering has been a well-studied problem in computer graphics and CAD communities [4, 60, 2, 1, 38], enabling the recovery of a structured representation of shapes for applications such as editing and manufacturing. Earlier works focused on reconstructing CAD through inverse CSG modeling [9, 55, 22, 48, 69], which decomposes shapes into simple primitives combined with boolean operations. While achieving good reconstruction, CSG-like methods tend to combine a large number of shape primitives, making them less flexible or easily manipulated by users. Recent efforts in collating large-scale CAD datasets (ABC [25], Fusion 360 [64], SketchGraphs [54]) have pushed this boundary, enabling the learning of data-driven methods for various applications such as B-Rep classification [19] and segmentation [19, 26, 10], parametric surface inference [16, 36], and CAD reconstruction [23, 27] and generation [66]. For the reverse engineering task, a common CAD modeling paradigm uses *sketch-extrude* [64, 58] as the basic building block, where a sketch [67, 47] is defined as a sequence of parametric curves forming a closed loop. This results in primitives that are more flexible and complex, not a predefined finite set, which will be the focus of this work.

Previous research uses NLP-based language models conditioned on raw geometry to represent CAD models as token sequences [65, 14, 18, 72]. However these methods are not designed to understand and decompose geometry into individual primitives and may also result in invalid CAD sequences. A recent work [21] also models CAD with language but face similar issues despite allowing multi-view inputs. To address these issues, Point2Cyl [58] is the pioneering work that poses the CAD reverse engineering task as a geometry-aware decomposition problem. The idea was to first decompose an input point cloud into *extrusion cylinders* (sketch-extrude primitives) and introduce differentiable loss functions to recover the parameters. ExtrudeNet [49] and SECAD-Net [30] further built on this setting and extended it to the unsupervised and self-supervised settings, respectively. Our work tackles a similar setting that takes a geometry-aware approach to the reverse engineering task, except we take multi-view images as input, as opposed to existing works that only handle point cloud inputs.

# 3 MV2Cyl: Reconstructing 3D Extrusion Cylinders from Multi-View Images

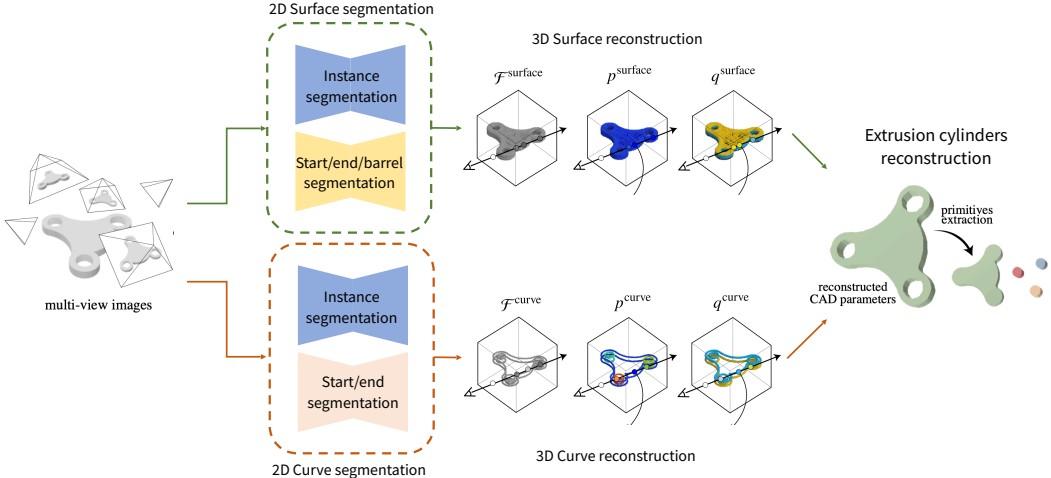

Figure 1: **Full Pipeline of MV2Cyl.**

We propose a method that reconstructs 3D extrusion cylinders from multi-view images without relying on raw 3D geometry as input. The idea is to leverage 2D neural networks to learn 2D priors that provide 3D extrusion information (Sec. 3.2), i.e. extrusion curves and surfaces. The integration of the information into 3D is achieved through optimizing neural fields (Sec. 3.3). MV2Cyl is a combination of a curve field and a surface field that is used to recover the parameters and reconstruct the extrusion cylinders only given 2D input images (Sec. 3.4).

## 3.1 Problem Statement and Background

Given a set of 2D RGB images $\{\mathbf{I}_i\}_{i=1}^{N}$ where $\mathbf{I} \in \mathbb{R}^{H \times W \times 3}$, the goal is to recover a set of extrusion cylinders $\{\mathbf{E}_j\}_{j=1}^{K}$ that represent the underlying 3D shape. We provide an overview here but refer the readers to Uy *et al.* [58] for more details.

**Sketch-Extrude CAD** is a designing paradigm or the designed model itself in the area of computer-aided design (CAD) consists of extrusion cylinders as its building blocks. An **extrusion cylinder** $\mathbf{E}$ is 3D space defined as $\mathbf{E} = (\mathbf{n}, \mathbf{c}, \mathbf{h}, \tilde{\mathbf{S}}, \mathbf{s})$ with the **extrusion axis** $\mathbf{n} \in \mathbb{S}^2$, the **extrusion center** $\mathbf{c} \in \mathbb{R}^3$, the **extrusion height** $\mathbf{h} \in \mathbb{R}$, and the normalized **sketch** $\tilde{\mathbf{S}}$ scaled by $\mathbf{s} \in \mathbb{R}$. A sketch-extrude CAD model is a 3D shape that is reconstructed from a set of extrusion cylinders, $\{E_1, E_2, ..., E_K\}$ where K is the number of extrusion cylinders. A **sketch** $\tilde{\mathbf{S}}$ is a set of non-self-intersecting closed loops drawn in a normalized plane.

We further classify the surfaces of a sketch-extrude CAD model as **base** or **barrel**. A surface is a **base** if its surface normal is parallel to its extrusion axis $\mathbf{n}$, and is a **barrel** if its surface normal is perpendicular to its extrusion axis. By this definition, base surfaces are parameterized by points on the planes and a normal ($\mathbf{n}$). The base surfaces can further be distinguished into start plane and end plane where the **start plane** contains the point $\mathbf{c} - \frac{\mathbf{h}}{2}\mathbf{n}$, while the **end plane** contains the point $\mathbf{c} + \frac{\mathbf{h}}{2}\mathbf{n}$.

With these definitions, we will later show how our method, MV2Cyl, is able to recover the set of extrusion cylinders and their corresponding parameters given only 2D multi-view images.

## 3.2 Learning 2D Priors for 3D Extrusions

To reconstruct extrusion cylinders from multi-view images, we first learn 2D priors for 3D extrusions by exploiting 2D convolutional neural networks. We train two U-Net-based 2D segmentation frameworks: $\mathcal{M}^{\text{curve}}$ that extracts curve information and $\mathcal{M}^{\text{surface}}$ that extracts surface information. The 2D information extracted from the two frameworks is integrated into 3D using a neural field and then utilized for more robust reverse engineering. We detail each 2D segmentation frameworks below.

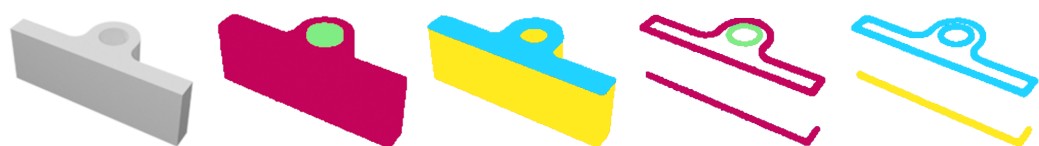

Figure 2: **Example of segmentation prediction.** From left to right: input rendered image, surface instance segmentation, surface start-end-barrel segmentation, curve instance segmentation, and curve start-end segmentation.

**2D Surface Segmentation Framework** $\mathcal{M}^{\text{surface}}$. The goal of the surface segmentation network is to extract the extrusion instance segmentation as well as start plane, end plane, and barrel (start-end-barrel) segmentation given an input image. That is $\mathcal{M}^{\text{surface}} : \mathbb{R}^{H \times W \times 3} \to \{\mathbf{P}^{\text{surface}} \in \{0, ..., K\}^{H \times W}, \mathbf{Q}^{\text{surface}} \in \{0, 1, 2, 3\}^{H \times W}\}$, where $K$ is the number of instances, $\mathbf{P}^{\text{surface}}$ represents the extrusion instance segmentation label, and $\mathbf{Q}^{\text{surface}}$ denotes the start-end-barrel segmentation label. The zero index is used for background annotation. $\mathcal{M}^{\text{surface}}$ is implemented as two distinct U-Nets predicting each segmentation and is trained using a multi-class cross entropy loss with ground truth labels $\hat{\mathbf{P}}^{\text{surface}}$ and $\hat{\mathbf{Q}}^{\text{surface}}$. An important property is that the problem does not admit a unique solution as i) the extrusion segment can be ordered arbitrarily and ii) the start and end planes can also be arbitrarily labeled. To handle this, we use Hungarian matching to find the best one-to-one matching with the ground truth labels and reorder it as $\tilde{\mathbf{P}}^{\text{surface}}$ and $\tilde{\mathbf{Q}}^{\text{surface}}$. This gives us the loss function of the instance segmentation:

$$\mathcal{L}_{\text{2D}}^{\text{surface}} = -\frac{1}{BHW} \sum_{b=1}^{B} \sum_{h=1}^{H} \sum_{w=1}^{W} \sum_{k=0}^{K} \mathbb{1}[\tilde{\mathbf{P}}_{hw}^{\text{surface}} = k] \log \mathbf{P}_{hwk}^{\text{surface}}, \quad (1)$$

where $\mathbf{P}_{hwk}^{\text{surface}}$ is the model's predicted probability that pixel $(h, w)$ belongs to the $k$-th instance in the $b$-th image in the batch, and $\tilde{\mathbf{P}}_{hw}^{\text{surface}}$ is the pseudo GT label for that pixel, with the number of images in a batch $B$ and the number of possible instances including the background $K$. The loss function of the start-end-barrel segmentation appears the similar, replacing the $\mathbf{P}$ as $\mathbf{Q}$ and the range of classes $\{0, ..., K\}$ to $\{0, 1, 2, 3\}$.

**2D Curve Segmentation Framework** $\mathcal{M}^{\mathbf{curve}}$. We observe that detecting and extracting curves on the base planes of each extrusion cylinder segment provides additional information to recover the parameters and reconstruct the extrusion cylinders. Moreover, (feature) curves have been a longstanding and well-explored problem dating back to classical computer vision literature [46, 74, 43] thanks to their strong expressiveness. This leads us to learn a 2D curve prior that provides more discriminative and detailed outputs to make reverse engineering more robust.

The goal of the curve segmentation framework is to extract both extrusion instance segmentation and start-end segmentation, where the start-end plane segmentation distinguishes the curves of the two base planes of extrusion cylinders. Concretely, $\mathcal{M}^{\mathrm{curve}} : \mathbb{R}^{H \times W \times 3} \rightarrow \left\{ \mathbf{P}^{\mathrm{curve}} \in \{0, ..., K\}^{H \times W}, \mathbf{Q}^{\mathrm{curve}} \in \{0, 1, 2\}^{H \times W} \right\}$, where $\mathbf{P}^{\mathrm{curve}}$ is the extrusion instance segmentation label and $\mathbf{Q}^{\mathrm{curve}}$ is the start-end segmentation label.

Similar to $\mathcal{M}^{\mathrm{surface}}$, the zero index is used for background annotation in both segmentation tasks and $\mathcal{M}^{\mathrm{curve}}$ is also implemented with two U-Nets and trained using a multi-class cross-entropy loss against the pseudo ground truth $\left\{ \tilde{\mathbf{P}}^{\mathrm{curve}}, \tilde{\mathbf{Q}}^{\mathrm{curve}} \right\}$ that is reordered with Hungarian matching to handle order invariance and ambiguities. An additional challenge for curves compared to surfaces is the labels are a lot sparser with the majority of the images being background pixels. Hence, the model can easily fail to predict meaningful labels due to this label imbalance. To alleviate this, we additionally employ a dice loss [41] to handle the strong foreground-background imbalance and a focal loss [33] to circumvent the class imbalance between the extrusion instances. Hence the loss function used to train the curve prior is given as:

$$\mathcal{L}_{\mathrm{2D}}^{\mathrm{curve}} = \lambda_{CE} \mathcal{L}_{\text{cross-entropy}} + \lambda_{\mathrm{focal}} \mathcal{L}_{\mathrm{focal}} + \lambda_{\mathrm{dice}} \mathcal{L}_{\mathrm{dice}}. \tag{2}$$

Additional details about the segmentation framework are available in the appendix E.1.

## 3.3 Integrating 2D Segments into a 3D Field

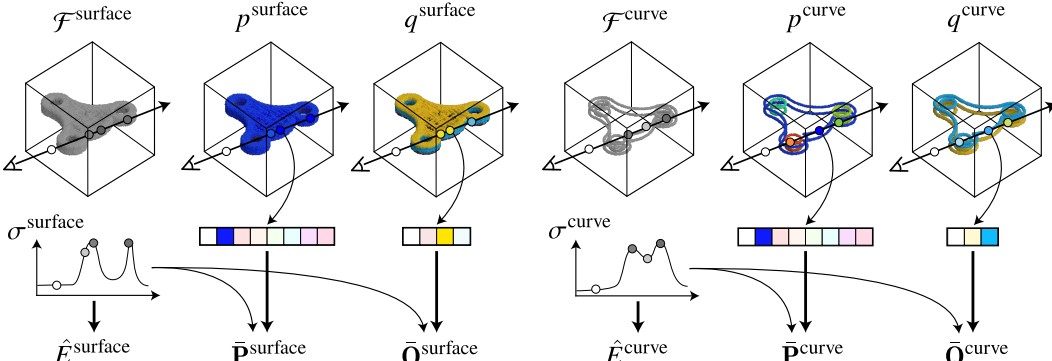

Figure 3: **Overview of the learned surface and curve fields.** (Left-to-Right) Density field of surface, instance semantic field of surface, start-end semantic field of surface, density field of curve, instance semantic field of curve, and start-end semantic field of curve.

To establish correspondences and collate the information extracted from multi-view images, a natural approach to *integrate* 2D information into a consistent 3D representation is through learning a 3D field [40, 61, 68, 44]. For each of our 2D priors $\mathcal{M}^{\mathrm{surface}}$ and $\mathcal{M}^{\mathrm{curve}}$, we learn a density field $\mathcal{F}$ and a semantic field $\mathcal{A}$ as detailed below, respectively.

**Density Field** $\mathcal{F}$. We learn a density field for surfaces $\mathcal{F}^{\mathrm{surface}}$ and curves $\mathcal{F}^{\mathrm{curve}}$. Given a query point in 3D space $\mathbf{x} \in \mathbb{R}^3$, the density field network $\mathcal{F} : \mathbb{R}^3 \mapsto \mathbb{R}$ outputs a scalar value that indicates how likely the 3D point $\mathbf{x}$ is on a surface or a curve for $\mathcal{F}^{\mathrm{surface}}$ and curves $\mathcal{F}^{\mathrm{curve}}$, respectively. To optimize the field differentiably with only 2D images, we use volume rendering [40, 5] and a learnable transformation [68] to convert the density field $\mathcal{F}(\mathbf{x})$ to an opacity field $\sigma(\mathbf{x})$. This transformation is in the form of a learnable sigmoid function given as:

$$\sigma(\mathbf{x}) = \alpha \cdot \left( 1 + e^{-\zeta \cdot (\mathcal{F}(\mathbf{x}) - \beta)} \right)^{-1} \tag{3}$$

where $\alpha$ is a learnable parameter that adjusts the density scale, and $\beta$ and $\zeta$ are constant hyperparameters. We use the same transformation in Eq. 3 for the surface field $\mathcal{F}^{\text{surface}}$ and the curve field $\mathcal{F}^{\text{curve}}$, resulting in corresponding opacity fields $\sigma^{\text{surface}}$ and $\sigma^{\text{curve}}$. We then directly render the density by using the volume rendering equation to get the expected surface and curve density for a camera ray $\mathbf{r}$, which is given by the integrals

$$\hat{E}^{\text{surface}}(\mathbf{r}) = \int_{t_n}^{t_f} T^{\text{surface}}(t)\sigma^{\text{surface}}(\mathbf{r}(t))dt, \tag{4}$$

$$\hat{E}^{\text{curve}}(\mathbf{r}) = \int_{t_n}^{t_f} T^{\text{curve}}(t)\sigma^{\text{curve}}(\mathbf{r}(t))dt, \tag{5}$$

where $T^{\text{surface}}(t) = \int_0^t \sigma^{\text{surface}}(\mathbf{r}(s))ds$ and $T^{\text{curve}}(t) = \int_0^t \sigma^{\text{curve}}(\mathbf{r}(s))ds$ are the corresponding transmittance. Inspired by NEF [68], we train our surface field $\mathcal{F}^{\text{surface}}$ and curve field $\mathcal{F}^{\text{curve}}$ using an adaptive L2 loss for the reconstruction of pixel density and a sparsity loss for learning of a sharp distribution of density along the ray. The objective function is given as follows:

$$\begin{aligned} L_{\text{density}} = &\frac{1}{N_{\text{rays}}} \sum_{\mathbf{r}} W_{\text{batch}}(\mathbf{r})||E(\mathbf{r}) - \hat{E}(\mathbf{r})||_2^2 \\ &+ \lambda_{sparsity} \frac{1}{N_{\text{samples}}} \sum_{i,j}(\mathbf{r}) \log\left(1 + \frac{\mathcal{F}(\mathbf{r}_i(t)_j)^2}{s}\right), \end{aligned} \tag{6}$$

where $E(\mathbf{r})$ and $\hat{E}(\mathbf{r})$ are the 2D-observed and predicted field density, respectively, for both the surface and curve models, $i$ indexes background pixels from input maps, $j$ indexes the sample points along the rays, and $s$ determines the scale of the regularizer. The 2D-observed density $E(\mathbf{r})$ is derived from the output of the 2D prior being a foreground pixel, either start, end, or barrel pixel.

Following Ye *et al.* [68], $W_{\text{batch}}(\mathbf{r})$ is given as:

$$W_{\text{batch}}(\mathbf{r}) = \begin{cases} \frac{N_{\text{batch exist pixels}}}{N_{\text{batch pixels}}} & \text{if } \mathbf{r} > \eta_{\text{batch}} \\ 1 - \frac{N_{\text{batch exist pixels}}}{N_{\text{batch pixels}}} & \text{otherwise} \end{cases} \tag{7}$$

The weight $W_{\text{batch}}(r)$ addresses the imbalance between foreground and background areas in the density maps, preventing the network from converging to a local minimum that falsely classifies most pixels as background, which are more prevalent.

**Semantic Field $\mathcal{A}$.** Now with the density field, we can distill the information from the 2D priors by learning semantic fields $\mathcal{A}$ for both surfaces and curves, which we denote as $\mathcal{A}^{\text{surface}}$ and $\mathcal{A}^{\text{curve}}$. Following Zhi *et al.* [73], each semantic is modeled as an additional feature channel branching out from the density field $\mathcal{F}$. For surfaces, the additional semantic field $\mathcal{A}^{\text{surface}} = \{p^{\text{surface}}, q^{\text{surface}}\}$ is comprised of two networks $p^{\text{surface}} : \mathbb{R}^3 \mapsto \mathbb{R}^{(K+1)}$ that predicts the extrusion instance label, and $q^{\text{surface}} : \mathbb{R}^3 \mapsto \mathbb{R}^4$ that predicts the start-end-barrel segmentation. Similarly for the curves, the semantic field is given as $\mathcal{A}^{\text{curve}} = \{p^{\text{curve}}, q^{\text{curve}}\}$, where $p^{\text{curve}} : \mathbb{R}^3 \mapsto \mathbb{R}^{(K+1)}$ predicts the extrusion instance label, and $q^{\text{curve}} : \mathbb{R}^3 \mapsto \mathbb{R}^3$ predicts the start-end segmentation. The expected semantic for a ray $\mathbf{r}$ is then computed using the volume rendering equation which is given as:

$$\hat{A}(\mathbf{r}) = \int_{t_n}^{t_f} T(t)\sigma(\mathbf{r}(t))a(\mathbf{r}(t))dt, \tag{8}$$

for each $a \in \{p^{\text{surface}}, q^{\text{surface}}, p^{\text{curve}}, q^{\text{curve}}\}$ that correspond to rendered predicted semantics $\hat{A} \in \{\bar{\mathbf{P}}^{\text{surface}}, \bar{\mathbf{Q}}^{\text{surface}}, \bar{\mathbf{P}}^{\text{curve}}, \bar{\mathbf{Q}}^{\text{curve}}\}$ from the 3D field.

We train our semantic field using the predictions from our learned 2D priors $\mathcal{F}^{\text{surface}}$ and $\mathcal{F}^{\text{curve}}$. For each of the 2D multi-view images, we can extract the 2D observed labels $A \in \{\mathbf{P}^{\text{surface}}, \mathbf{Q}^{\text{surface}}, \mathbf{P}^{\text{curve}}, \mathbf{Q}^{\text{curve}}\}$ from the learned priors, which we use as supervision. As mentioned in Sec. 3.2, since the labels may not individually be consistent across the multiple views, we use Hungarian matching during training to align the labels across multiple views based on the best one-to-one matching. For each semantic, we use a multi-class cross-entropy loss to train and optimize the semantic field with the given supervision:

$$L_{semantic} = -\sum_{\mathbf{r}} \sum_{l=0}^{L} A^l(\mathbf{r}) \log \hat{A}^l(\mathbf{r}), \tag{9}$$

where $A^l(\mathbf{r})$ and $\hat{A}^l(\mathbf{r})$ represent the class $l$ semantic probabilities from the learned 2D prior and the predicted field, respectively, aligned with Hungarian matching.

**Training and Implementation Details**. The integration into a coherent 3D field is optimized independently for each 3D shape. We use TensoRF [5] as the backbone representation for the 3D fields, where we have one model for the surface field and the other for the curve field. In both models, we train the density fields $\mathcal{F}$ and the semantic fields $\mathcal{A}$ together for 1500 iterations. Each semantic field is parameterized by a 2-layer MLP that outputs logit values. Additional implementation details are provided in the Appendix E.2.

### 3.4 Reverse Engineering from 3D Fields

We now elaborate on how we leverage our optimized surface field $\mathcal{F}^{\text{surface}}$ and curve field $\mathcal{F}^{\text{curve}}$ for the reverse engineering task, specifically to recover extrusion cylinders and their defining parameters, $(\mathbf{n}, \mathbf{c}, \mathbf{h}, \tilde{\mathbf{S}}, \mathbf{s})$. Each point within the extracted point clouds is assigned semantic information queried from the associated semantic fields, providing context for each feature in 3D space. Using these enriched point clouds, we employ a multi-step process to reconstruct each extrusion cylinder, iteratively refining the parameters to accurately fit the geometry and semantics. Further details can be found in Appendix E.3.

1. **Extrusion Axis n Estimation**: For each extrusion cylinder, the extrusion axis $\mathbf{n}$ is computed through plane fitting via RANSAC [12] on the surface base points with corresponding extrusion cylinder instance label. The axis $\mathbf{n}$ is then set as the normal of the fitted plane. This process relies on surface information because plane fitting is more reliable using the surface points. If the base face of the cylinder fails to be reconstructed due to occlusion, we use curve points as a substitute.

2. **2D Sketch $\tilde{\mathbf{S}}$ Estimation**: To obtain the sketch $\tilde{\mathbf{S}}$ for each extrusion cylinder, we first project the curve point cloud for each extrusion instance onto the plane with normal $\mathbf{n}$. From the projected point cloud, we then compute for the sketch scale $\mathbf{s}$. We then optimize an implicit signed distance function for each closed loop using the projected point cloud with IGR [15]. Finally, we obtain the closed loop by extracting the zero-level set of the implicit function. Implicit sketches can be converted to a parametric representation using an off-the-shelf spline fitting module. We provide example visualizations in Appendix E.7.

3. **Extrusion height h Estimation**: The height $\mathbf{h}$ of each extrusion cylinder is computed as the distance between the two sketch centers — from the start plane and the end plane. The sketch centers are computed from the projected curve point cloud for each paired labels (extrusion instance segmentation, start-end segmentation) onto the plane with normal $\mathbf{n}$. If any of the start-end planes are occluded, the height is computed using the barrel points from the surface point cloud with the corresponding extrusion cylinder instance label.

4. **Extrusion centroid c Estimation**: The centroid $\mathbf{c}$ of each extrusion cylinder can be obtained by taking the average position of the 3D curve point cloud with the corresponding extrusion cylinder instance label. Similar to the above, if any of the start-end planes are occluded or missing, the centroid is computed as the average of the barrel points from the surface point cloud with the corresponding extrusion cylinder instance label.

## 4 Experiments

In this section, we present our experimental evaluation to demonstrate the feasibility of our MV2Cyl in reconstructing extrusion cylinders given only multi-view images. To the best of our knowledge, our proposed pipeline is the first to tackle extrusion cylinder recovery directly from 2D input images, and we show that our method is outperforms existing methods that take clean geometry as input and their modifications to handle raw input images.

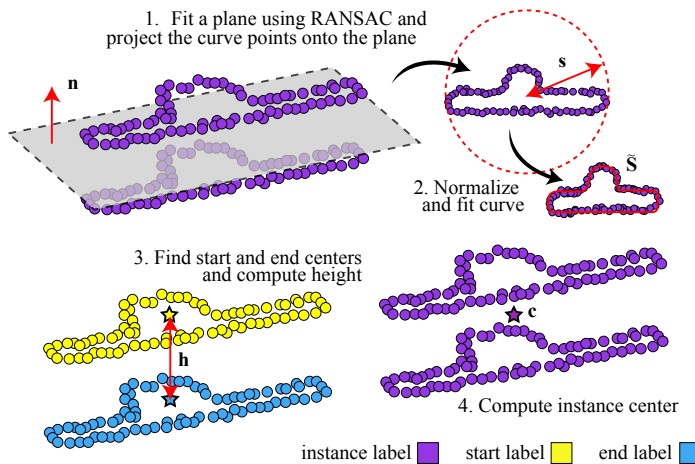

Figure 4: Converting 3D reconstructed geometry and semantics into CAD parameters.

**Datasets**. We evaluate our approach on two sketch-extrude CAD datasets Fusion360 [64] and DeepCAD [65]. We use the train and test splits as released in [58]. We enrich these datasets with multi-view image renderings and 2D curve and surface segmentation maps for the training and evaluation of our method and the baselines. More details about dataset generation are in Appendix E.4.

**Evaluation Metrics**. Following [58], we report the average extrusion-axis error (E.A.), extrusion-center error (E.C.), per-extrusion cylinder fitting loss (Fit Cyl.), and global fitting loss (Fit Glob.). Additionally, we also report the extrusion-height error (E.H.) defined as the L1 distance between the GT and predicted height value: $\frac{1}{K}\sum_{k=1}^{K}|\hat{\mathbf{h}}_k - \mathbf{h}_k|$. These metrics evaluate how well the methods reconstruct and recover the parameters of the extrusion cylinders. The final scores are the average of the metrics calculated over all test shapes.

**Baselines**. In the absence of established methods for transforming multi-view images into sketch-extrude CAD models within shape space, we benchmark our model against a point cloud to sketch-extrude CAD reconstruction technique [58] and its variant to validate the effectiveness of our proposed approach.

- We compare with **Point2Cyl [58]**, a technique for reconstructing sketch-extrude CAD given input 3D point clouds. The point clouds are sampled from clean ground truth CAD meshes and are utilized as the input for this baseline. Point2Cyl [58] initially segments the input point cloud into distinct instances and base/barrel segments, then identifies the extrusion axis as the direction that aligns with the normals of the points in the base segments and is orthogonal to the normals of the points in the barrel segments. The rest of the parameters are inferred based on this predicted axis and the segmentation labels. To evaluate the network, we trained it using the official code released by the authors.

- **NeuS2 [62]+Point2Cyl [58]** combines NeuS2 [62] with Point2Cyl [58]. Since no prior work has reconstructed extrusion cylinders directly from multi-view images, we combine methods for i) shape reconstruction from multi-view images, with techniques for ii) extrusion cylinder reconstruction from raw geometry. This approach utilizes NeuS2 [62], an off-the-shelf image-based surface reconstruction technique, that optimizes the photometric loss to learn underlying implicit 3D representations. For this baseline, we first reconstruct the object surface from multi-view images using NeuS2 [62], and then sample a point cloud from the reconstructed mesh. The sampled point cloud is then directly fed into Point2Cyl [58] to predict sketches and extrusion parameters of CAD models.

## 4.1 Results

Tab. 1 shows the quantitative results of CAD reconstruction on the Fusion360 [64] and DeepCAD [65] datasets. We see that MV2Cyl outperforms the baselines by considerable margins across both datasets in all evaluated metrics. Qualitative comparisons are shown in Fig. 5.

Table 1: **Quantitative results using Fusion360 [64] and DeepCAD [65] datasets.** MV2Cyl consistently outperforms all baselines in all metrics. Red denotes the best; orange denotes the second best.

| Dataset | Method | E.A.(°)↓ | E.C.↓ | E.H.↓ | Fit. Cyl. ↓ | Fit. Glob.↓ |
|---------|--------|----------|-------|-------|-------------|-------------|
| Fusion 360 | NeuS2+Point2Cyl | 35.0562 | 0.2198 | 0.7802 | 0.1036 | 0.0596 |
| | Point2Cyl | 9.5228 | 0.0839 | 0.2918 | 0.0731 | 0.0293 |
| | **MV2Cyl (Ours)** | 1.3939 | 0.0385 | 0.1423 | 0.0284 | 0.0212 |
| Deep CAD | NeuS2+Point2Cyl | 54.8715 | 0.0650 | 0.8471 | 0.0903 | 0.0638 |
| | Point2Cyl | 7.9156 | 0.0266 | 0.1632 | 0.0420 | 0.0250 |
| | **MV2Cyl (Ours)** | 0.2202 | 0.0121 | 0.0761 | 0.0246 | 0.0229 |

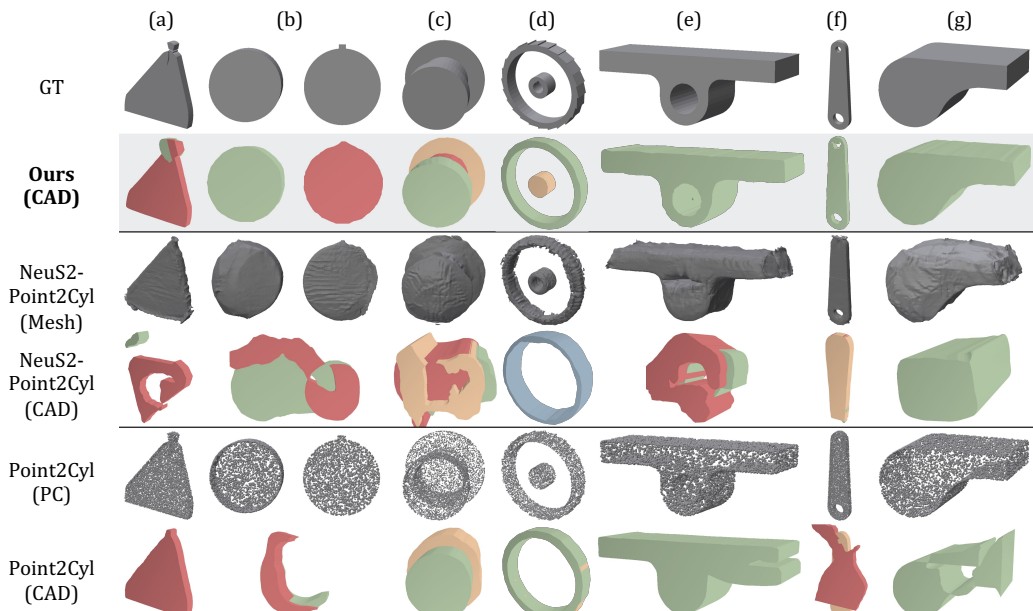

Figure 5: **Qualitative comparisons with the baselines.** Each instance is identified by a different color. MV2Cyl produces high-quality geometry and even outperforms Point2Cyl [58] that directly consumes 3D point clouds. Furthermore, the comparison against a naive baseline that pipelines NeuS2 [62], a multi-view surface reconstruction technique, to Point2Cyl demonstrates the importance of edge information when inferring 3D structures.

The results of NeuS2 [62]+Point2Cyl [58] illustrate the challenges in naively using multi-view images for CAD reconstruction, where inaccurate 3D geometry reconstructed from images serve as a critical bottleneck in recovering CAD parameters. We observe that it struggles to distinguish the object's intrinsic color (albedo) from the effects of lighting or shadows (shading). This can result in discrepancies, such as omitted details or subtle smoothing at the intersection between different extrusion segments, as illustrated in (Fig. 5). As seen in Fig. 5 (c), a common failure arises when NeuS2 [62] fails to reconstruct fine geometric details, in this case, the boundaries between the extrusion cylinders. Such artifacts in the reconstructed mesh make the conversion from the point cloud to the CAD model unreliable.

MV2Cyl also outperforms Point2Cyl [58], which takes precise 3D geometric information as input. An advantage of taking multi-view images instead of point clouds is the images contain richer information such as edges and curves. Moreover, 2D neural networks have also shown superior performance compared to 3D processing networks leading to superior performance of MV2Cyl compared to Point2Cyl [58]. The comparison of the 2D segmentation network in MV2Cyl with the 3D segmentation network in Point2Cyl are found in Appendix F.1.

Unlike the baselines that require accurate 3D surface information, MV2Cyl is able to directly handle 2D multi-view images as input by leveraging on learned 2D priors to extract 3D extrusion information. We show that extracting extrusion surfaces and curves from 2D images allows us to leverage both the curve and surface representation for better extrusion cylinder reconstruction compared to baselines that even require clean 3D geometry as input.

### 4.2 Real Object Reconstruction

We validate MV2Cyl on a real-world demo by 3D-printing various sketch-extrude objects from the Fusion360 test set, by capturing multi-view images with an iPhone 12 Mini and then obtaining camera poses via COLMAP. These captured images were processed through our pipeline to extract 2D information, converted to 3D using our density and semantic fields to recover extrusion cylinders and parameters. To address the domain gap between synthetic training images and real-world data, we applied preprocessing techniques, including grayscale conversion, background removal, and then fine-tuning a large vision model. Additional details are provided in Appendix B.

## 5    Conclusion

We introduce MV2Cyl, a novel method for reconstructing a set of extrusion cylinders in 3D from multi-view images. Unlike previous approaches that rely only on raw 3D geometry as input, our framework utilized 2D multi-view images and demonstrates superior performance by leveraging the capabilities of 2D convolutional neural networks in image segmentation. Through the analysis of curve and surface representations, each having its own advantages and shortcomings, we propose an integration of both that takes the best of both worlds, achieving optimal performance.

**Limitations and Future Work**. Our proposed method is not without its limitations. While MV2Cyl demonstrates state-of-the-art performance in reconstructing sketch-extrude CAD models, it shares a limitation with previous approaches, such as Point2Cyl [58], regarding the prediction of binary operations among primitives. MV2Cyl does not explicitly predict these binary operations, which we leave as future work. Nonetheless, we propose an initial straightforward method for recovering binary operations through exhaustive search. Details are provided in the appendix E.6. Additionally, MV2Cyl does not directly accommodate textured CAD models, as the 2D segmentation models are trained on a synthetic dataset derived from rendering untextured CAD models. By utilizing generalizable large 2D models, such as Segment Anything [24], we can preprocess images of textured CAD models for segmentation by extracting object masks and removing textures. We also leave this for future investigation.

MV2Cyl is also susceptible to occlusion since it relies on 2D image inputs. Specifically, if one side of an extrusion cylinder is completely hidden by others, our 2D segmentation model cannot detect the hidden side. In such cases, the extrusion cylinder cannot be reconstructed. In Fig. 6, the left shape represents the target CAD model, while the right one shows the reconstructed model by MV2Cyl. The target shape features an inset hexagonal cylinder with one end hidden by an outer cylinder, but MV2Cyl was unable to reconstruct the inset extrusion cylinder.

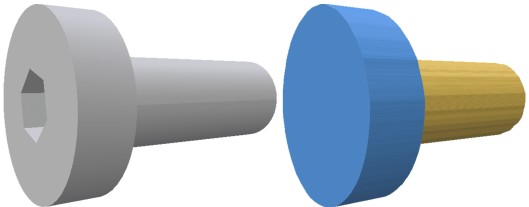

Figure 6: **Example of a failure case.**

**Societal Impacts**. This technology could make CAD more accessible to non-professionals or educational sectors, allowing more people to engage in design and engineering tasks without needing extensive training in traditional CAD software. However, such a 3D reconstruction method can also raise privacy concerns. By converting a physical object into a digital format, the shape could be reused for design purposes without the original creator's permission.

## Acknowledgements

We thank Seungwoo, Juil, Jaihoon, and Yuseung for their valuable comments and suggestions in developing the idea and preparing an earlier version of the manuscript. This work was supported by the NRF grant (RS-2023-00209723), IITP grants (RS-2022-II220594, RS-2023-00227592, RS-2024-00399817), and KEIT grant (RS-2024-00423625), all funded by the Korean government (MSIT and MOTIE), as well as grants from the DRB-KAIST SketchTheFuture Research Center, NAVER-Intel Co-Lab, Hyundai NGV, KT, and Samsung Electronics. Mikaela also acknowledges the support from an Apple AI/ML PhD Fellowship.

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

# Appendix

## A    Ablation Study

### A.1    Importance of Jointly Using Surface and Curve

Tab. A2 shows the importance of combining surface and curve reconstruction for CAD reconstruction. Fig. A7 shows qualitative results of the final CAD model outputs and also includes intermediate results, such as point clouds from density fields.

We see that in surface-only reconstruction, occlusion between instances can result in missing base faces, as seen with the central cylinder in the top row of Fig. A7 (a). This occlusion impedes accurate CAD parameter reconstruction. Conversely, the curves (see Fig. A7 (a)) capture this extrusion instance with accurate start-end labels, enabling successful CAD reconstruction as demonstrated in the curve only and MV2Cyl CAD outputs.

In Fig. A7 (b), we see that the curve-only reconstruction struggles at the (small) intersection where the thin rod meet, failing to capture one base curve. This omission leads to a predicted zero height of the instance, highlighting a limitation in reconstructing that specific section using curve data alone. On the other hand, the surface reconstruction successfully reconstructs this shape. We leverage the reconstructed barrel face of the instance to accurately predict its height ($h$) and center ($c$). This approach enables MV2Cyl to successfully reconstruct instances that the curve only model may miss.

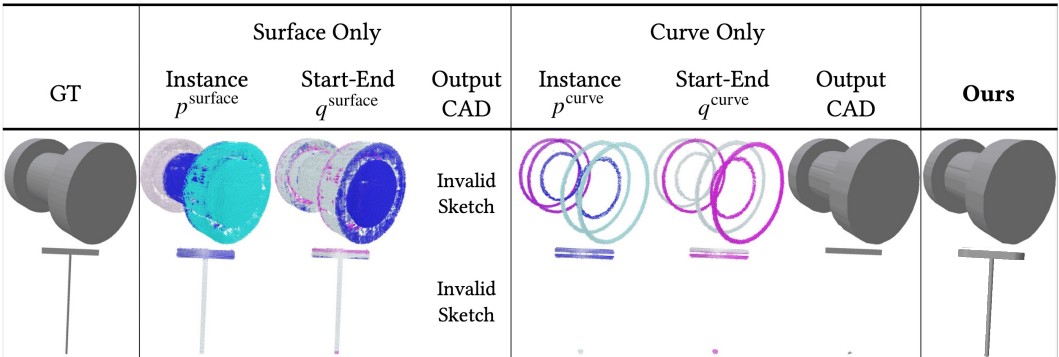

Figure A7: **Ablation study.** Each example illustrates the limitations of surface and curve representations. The surface-only model, whose instance label point cloud, start-end label point cloud, and output CAD model are shown in columns 2, 3, and 4, The table shows the final output of the surface-only model and the curve-only model along with their semantic fields. The surface-only model failed to reconstruct the proper start-end field and thus the sketch is invalid. The curve-only model was able to reconstruct the geometry in case (a) but predicted zero height for the thin cylinder in (b). Ours were able to make better estimations in both cases.

Table A2: **Quantitative results from the ablation study on Fusion360 [64] dataset.** The omission of curve fields leads to significant performance degradation (the first row). The performance could be further improved by joint use of surface reconstruction. Red denotes the best; orange denotes the second best.

| Method | E.A.(°)↓ | E.C.↓ | E.H.↓ | Fit. Cyl.↓ | Fit. Glob.↓ |
|---|---|---|---|---|---|
| Surface Only | 4.1180 | 0.1179 | 0.2376 | 0.0984 | 0.0353 |
| Curve Only | 1.5612 | 0.0434 | 0.1508 | 0.0310 | 0.0232 |
| Ours (Surface + Curve) | 1.3939 | 0.0385 | 0.1423 | 0.0284 | 0.0212 |

To summarize, our synergy of utilizing both surface and curve information allows MV2Cyl to effectively reconstruct instances even where individual data types may fail, showcasing its comprehensive

reconstruction capability. Specifically, curve representation is effective against occlusion and precisely captures the closed curve of a sketch, making it suitable for reconstructing the 2D implicit sketch. However, it may not always be reconstructed successfully due to its localized presence within the scene. On the other hand, surface representation provides a more dependable basis for fitting a 3D plane to determine the extrusion axis, which is vital for inferring subsequent CAD parameters. Yet, the base faces necessary for this process might not be available. This is either because of subtractive boolean operations or occlusion from combining multiple shapes. We thus address the challenges by jointly reconstructing curves and surface representations of shape and utilizing them to recover CAD parameters.

## A.2 Separate 2D Segmentation Modules for Curve and Surface

In MV2Cyl, the curve and surface segmentation modules are kept separate. To evaluate how segmentation performance changes when the 2D curve and surface segmentation networks are combined, we test a shared U-Net backbone that branches into two outputs for curve and surface segmentation. The results of using merged and separate U-Nets to extract 2D segmentation maps for our reverse engineering extrusion cylinders task are shown in Tab. A3. While a slight improvement in performance is observed on some metrics (Fit Cyl. and Fit Glob.), the results from both approaches are comparable.

Table A3: **Quantitative comparison with the U-Nets of surface and curve segmentation** using Fusion360 dataset.

| Method | E.A.(°)↓ | E.C.↓ | E.H.↓ | Fit. Cyl.↓ | Fit. Glob.↓ |
|---|---|---|---|---|---|
| Combined U-Net | 1.4425 | 0.0397 | 0.1543 | 0.0279 | 0.0200 |
| Separate U-Nets (**Ours**) | 1.3939 | 0.0385 | 0.1423 | 0.0284 | 0.0212 |

## A.3 Ablation on the Number of Instances

We evaluate the impact of varying the number of instances on MV2Cyl's performance using the Fusion360 dataset. Tab. A4 shows the results breakdown of the models in the Fusion360 test set across different numbers of extrusion instances K. Note that the number of samples for K = 7 and K = 8 is very small, and these statistics should therefore be interpreted sparingly.

Table A4: **Performance over the changes of the number of instances** using Fusion360 dataset.

| Number of instances | Number of samples | E.A.(°)↓ | E.C.↓ | E.H.↓ | Fit. Cyl.↓ | Fit. Glob.↓ |
|---|---|---|---|---|---|---|
| 1 | 250 | 0.0000 | 0.0011 | 0.0615 | 0.0169 | 0.0169 |
| 2 | 494 | 1.4484 | 0.0325 | 0.1531 | 0.0277 | 0.0224 |
| 3 | 207 | 2.9046 | 0.0732 | 0.2092 | 0.0405 | 0.0261 |
| 4 | 93 | 1.8476 | 0.1092 | 0.2270 | 0.0466 | 0.0300 |
| 5 | 40 | 2.4250 | 0.0998 | 0.2065 | 0.0586 | 0.0294 |
| 6 | 15 | 9.0000 | 0.1301 | 0.1916 | 0.0601 | 0.0459 |
| 7 | 7 | 1.3265 | 0.0712 | 0.0620 | 0.0553 | 0.0364 |
| 8 | 2 | 16.2143 | 0.0130 | 0.0350 | 0.0250 | 0.0232 |

## A.4 Ablation on the Number of the Input Images

In this section, we analyze the effect of varying the number of input images on MV2Cyl's performance using the Fusion360 dataset. Tab. A5 shows that our approach is robust to changes in the number of input images, maintaining reasonable performance even with as few as 10 input views. Although performance slightly degrades as the number of input images decreases, MV2Cyl still produces effective reconstructions, demonstrating its adaptability to limited input data. As expected, the use of 50 input images yields the best overall results due to the richer information available for reconstruction.

Table A5: **Performance over the changes of the number of the input images** using Fusion360 dataset.

| Number of images | E.A.(°)↓ | E.C.↓ | E.H.↓ | Fit. Cyl.↓ | Fit. Glob.↓ |
|---|---|---|---|---|---|
| 50 (**Ours**) | 1.3939 | 0.0385 | 0.1423 | 0.0284 | 0.0212 |
| 25 | 1.4631 | 0.0421 | 0.1561 | 0.0293 | 0.0218 |
| 15 | 2.1548 | 0.0489 | 0.1784 | 0.0371 | 0.0259 |
| 10 | 2.2585 | 0.0573 | 0.2112 | 0.0515 | 0.0364 |

## A.5 Ablation on the Line Width of the Curve Segmentation Maps

In this section, we investigate the impact of varying the line width of curve segmentation maps on MV2Cyl's performance. Tab. A6 shows that our method is relatively robust to changes in line width, with performance remaining consistent across different widths. The results indicate that using 5 pixels (our default setting) achieves a good balance between accuracy and consistency. While slight variations in error metrics are observed when using line widths of 2.5 or 7.5 pixels, these changes are minimal, suggesting that MV2Cyl can handle a range of line width settings without significant performance loss.

Table A6: **Performance over the changes of the line width of the curve segmentation maps** using Fusion360 dataset.

| Line width | E.A.(°)↓ | E.C.↓ | E.H.↓ | Fit. Cyl.↓ | Fit. Glob.↓ |
|---|---|---|---|---|---|
| 5 pixels (**Ours**) | 1.3939 | 0.0385 | 0.1423 | 0.0284 | 0.0212 |
| 2.5 pixels | 1.4886 | 0.0392 | 0.1482 | 0.0291 | 0.0216 |
| 7.5 pixels | 1.4375 | 0.0410 | 0.1548 | 0.0307 | 0.0225 |

## B Real-World Demo

In this section, we showcase the feasibility of using MV2Cyl in reconstructing CAD models from real-world object captures. We provide the details of the process as follows.

To reverse engineer a real-world object, we begin by capturing videos of the target object using an iPhone 12 Mini. These videos are then converted into multi-view images through frame extraction, which are subsequently input into the image segmentation models. The training dataset for our segmentation models is generated by rendering untextured models in Blender, producing RGB images with a grayscale appearance against a transparent background. To address the domain gap between the training data and real-world demo images, we preprocess the real images by converting them to grayscale and removing the background. While these preprocessing steps help align the datasets, a residual domain gap remains, which may still impact segmentation performance on real images. To address this challenge and enhance segmentation efficacy, we employ a robust 2D segmentation model known as Segment Anything (SAM)[24]. By fine-tuning the pretrained SAM model using LoRA-based finetuning code[71], with the same training dataset applied to the U-Nets, we observe substantial improvements in segmentation performance on real images, even in the presence of distributional shifts. The finetuned SAM model leverages extensive prior knowledge gained from training on a significantly larger dataset, ensuring robust performance in 2D segmentation tasks. We acquire the camera's extrinsic and intrinsic parameters using COLMAP [52, 53].

Fig. A8 displays an example of a real image alongside the predicted segmentation maps from the finetuned SAM model. The processed multi-view images and their corresponding camera poses are then input into the reconstruction pipeline, which focuses on learning the density and semantic fields. Subsequently, CAD parameters are estimated based on this reconstruction. Tab. A7 presents results on final reconstructed CAD models in real captures, demonstrating successful output of CAD models from various real-world objects.

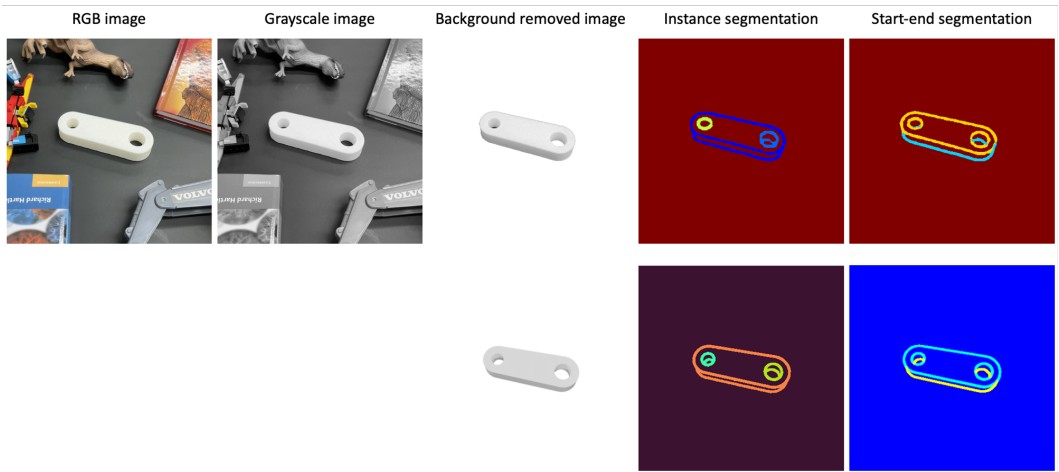

| RGB image | Grayscale image | Background removed image | Instance segmentation | Start-end segmentation |

Figure A8: **Example of the inferred segmentation maps** for a real image (top row) and a synthetic image (bottom row). The first image in the top row is the real image, which is then grayscaled (second image) and background-removed (third image). The fine-tuned SAM [24] model segments the processed image into the instance segment (fourth column) and the start-end segment (last column). When comparing the real and synthetic images from similar viewpoints, the segmentation outcomes are similarly effective.

We quantitatively evaluate MV2Cyl on real data found in Tab. A7. Our reconstruction achieves an average Chamfer distance of $2.11 \times 10^{-3}$ from the ground truth CAD model. To compute this distance, we first sampled 8192-point point clouds from the reconstructions and then aligned the reconstructed model to the corresponding ground truth shape using ICP. After alignment, the desired metrics were computed. This process is illustrated in Fig. A9.

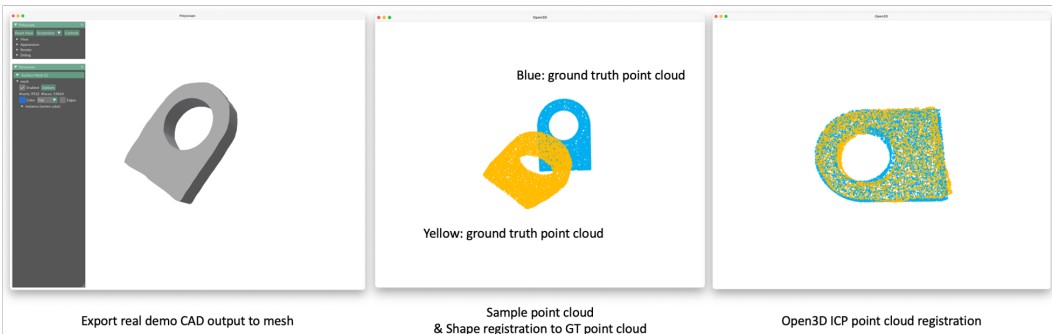

Figure A9: Processing steps for aligning the pose of the real demo output with the ground truth CAD point cloud.

## C    Comparison with SECAD-Net [30]

We further compare our MV2Cyl against an unsupervised CAD reconstruction baseline SECAD-Net [30]. Unlike our approach that takes multi-view images as input, SECAD-Net takes a 3D voxel grid as input. It trains a model that directly outputs a shape as a union of predicted extrusion cylinders. We evaluate how well their approach reconstructs individual extrusion cylinders using the same metrics as in [58] and compare them against our MV2Cyl. Results are shown in Tab. A8. We see that our approach outperforms SECAD-Net [30] for both the Fusion 360 [64] and DeepCAD [65] datasets on all metrics that quantify extrusion cylinder reconstruction.

For fair comparison, we evaluated the test set as released by [58] on both datasets and used the scripts from the code release of SECAD-Net [30] to convert the test shapes to voxelized inputs and to train or fine-tune the SECAD-Net model with our data. For both datasets, we adhered to the evaluation

Table A7: **Gallery of real demo examples.** It demonstrates the capabilities of MV2Cyl in reverse engineering real objects.

protocol from [30], fine-tuning SECAD-Net for 300 epochs per test shape using a pre-trained model, followed by mesh extraction at a 256 resolution with Marching Cubes [37]. We used Hungarian matching based on Chamfer distances to find the best match between ground-truth and predicted segments, which allowed us to calculate the metrics reported. The Fusion360 dataset evaluations were conducted on 798 test shapes, while the DeepCAD dataset included 1,950 test shapes. These numbers reflect instances where both MV2Cyl and SECAD-Net were able to produce valid CAD models.

Table A8: **Quantitative comparison with SECAD-Net [30] using Fusion360 [64] and Deep-CAD [65] datasets.** MV2Cyl consistently outperforms SECAD-Net [30] in all metrics. Red denotes the best.

| Dataset | Method | E.A.(°)↓ | E.C.↓ | E.H.↓ | Fit. Cyl. ↓ | Fit. Glob.↓ |
|---|---|---|---|---|---|---|
| Fusion 360 | SECAD | 42.5529 | 0.05881 | 0.6111 | 0.8378 | 0.7990 |
| | **MV2Cyl (Ours)** | 1.4654 | 0.04270 | 0.1520 | 0.0283 | 0.0215 |
| Deep CAD | SECAD | 43.2088 | 0.03406 | 0.5126 | 1.3627 | 0.6997 |
| | **MV2Cyl (Ours)** | 0.2476 | 0.00616 | 0.0917 | 0.0238 | 0.0227 |

Tab. A8 demonstrates that MV2Cyl outperforms SECAD-Net [30] across all metrics. SECAD-Net lacks supervision for segmenting parts in the way CAD designers would, leading to issues such as over-segmentation. This is evident in the images of Tab. A9. For example, in the second row, SECAD-Net segments two vertically attached cylinders into several patches along their surfaces. In comparison, MV2Cyl more accurately represents these components as two distinct cylinders, showing its improved ability to align with CAD design conventions.

Table A9: **Qualitative comparison with SECAD-Net [30] on Fusion360 [64] dataset.** MV2Cyl outperforms SECAD-Net for segment the instances.

| GT Mesh | Reconstruction | Extrusion Cylinders |
|---|---|---|
| | MV2Cyl | |
| | SECAD | |
| | MV2Cyl | |
| | SECAD | |
| | MV2Cyl | |
| | SECAD | |

# D Comparison with Curve + Point2Cyl [58]

We compare our MV2Cyl with an additional baseline Curve+Point2Cyl [58] that is a modified version of Point2Cyl that incorporates an additional "curve" feature into the input point cloud. While MV2Cyl benefits from the supervision of curve information from the CAD, we apply equivalent supervision to the baseline to maintain fairness in comparison. The evaluation was conducted using the Fusion360 [64] dataset.

To implement the baseline Curve+Point2Cyl (described in the Sec. 4 in the main paper), we modified the network's front-end where it receives inputs. The original Point2Cyl takes an input tensor of size $N \times 3$, representing the target point cloud (see Fig. A10). To incorporate information about specific curves within the point cloud, we concatenate the $M$ curve points after the $N$ input points. These curve points are assigned a label of one on an additional channel appended to the original three XYZ channels, while the original non-curve points retain a label of zero on this channel (see Fig. A11). The rest of the network architecture remains unchanged. $N$ is set to 8,192 and $M$ is set to 800. Notably, the curve points are obtained from the ground truth CAD models for both training and testing phases.

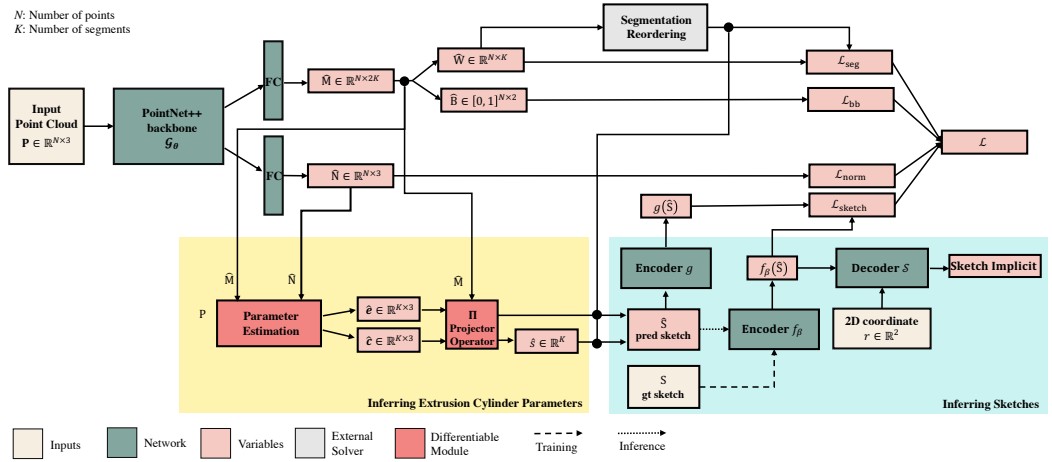

Figure A10: Network architecture of Point2Cyl [58]

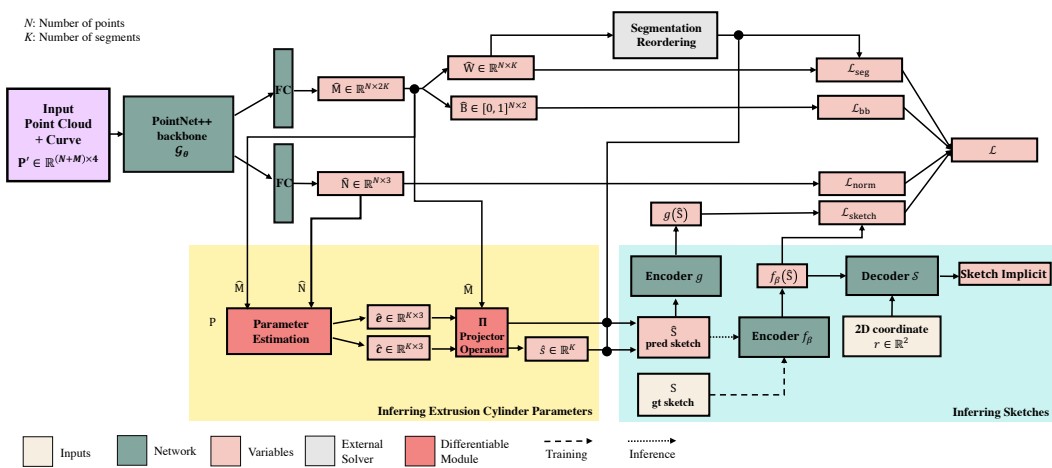

Figure A11: Network architecture of Curve+Point2Cyl [58]

Tab. A10 shows the quantitative comparison between MV2Cyl and Curve+Point2Cyl. MV2Cyl surpasses Curve+Point2Cyl across all the metrics. Despite incorporating ground truth curve features, Curve+Point2Cyl performs similarly to the original Point2Cyl. In practical applications, it might

be expected to use predicted curves or edges; however, even with ground truth curves, we observed no significant improvement over the original Point2Cyl, which lacks curve data. This suggests that Point2Cyl's network may prioritize surface points over boundary points, making the additional boundary information less impactful.

Table A10: **Quantitative comparison with Curve+Point2Cyl using Fusion360 [64] dataset.** MV2Cyl surpasses Curve+Point2Cyl across all the metrics. Red denotes the best.

| Method | E.A.(°)↓ | E.C.↓ | E.H.↓ | Fit. Cyl.↓ | Fit. Glob.↓ |
|---|---|---|---|---|---|
| Curve+Point2Cyl | 7.3253 | 0.0875 | 0.3001 | 0.0792 | 0.0327 |
| **MV2Cyl (Ours)** | 1.3939 | 0.0385 | 0.1423 | 0.0284 | 0.0212 |

# E  Implementation Details

In this section, we elaborate the implementation of the networks presented in the main paper. Sec. E.1 contains the details of the training 2D segmentation frameworks (Sec. 3.2 in the main paper) and Sec. E.2 includes the details of the 3D field learning from the 2D segments (Sec. 3.3 in the main paper).

## E.1  Details of 2D Segmentation Frameworks

Each 2D segmentation framework $\mathcal{M}^{\text{surface}}$ and $\mathcal{M}^{\text{curve}}$ includes two U-Nets for instance segmentation $\mathbf{P}^{\text{type}}$ and start-end(-barrel) segmentation $\mathbf{Q}^{\text{type}}$ where the type is either surface or curve, respectively.

**Details of 2D Surface Segmentation Framework $\mathcal{M}^{\text{surface}}$.**

The 2D surface segmentation framework, $\mathcal{M}^{\text{surface}}$, maps RGB images of dimensions $H \times W \times 3$ to a set comprising the extrusion instance segmentation label, $\mathbf{P}^{\text{surface}} \in \{0, \dots, K\}^{H \times W}$, and the start-end-barrel segmentation label $\mathbf{Q}^{\text{surface}} \in \{0, 1, 2, 3\}^{H \times W}$, where $K$ indicates the number of instances. We fix $K = 8$ instances following [58] and train with image resolutions of $H = W = 400$. For the Fusion360 dataset, 196,050 training images derived from 3,921 shapes are used, while for the DeepCAD [65] dataset, derived from a subset of the training set, 709,000 images from 14,180 shapes are utilized. The datasets are divided into training and validation sets in a 9:1 ratio for the U-Net [50] training. The training times for 3 epochs are 316 minutes for the Fusion360 [64] dataset and 381 minutes for the DeepCAD dataset, using a single NVIDIA RTX A6000 GPU.

**Details of 2D Curve Segmentation Framework $\mathcal{M}^{\text{curve}}$.**

The 2D curve segmentation framework, $\mathcal{M}^{\text{curve}}$, processes RGB images ($H \times W \times 3$) to produce two types of labels: the extrusion instance segmentation label, $\mathbf{P}^{\text{curve}} \in \{0, \dots, K\}^{H \times W}$, and the start-end segmentation label, $\mathbf{Q}^{\text{curve}} \in \{0, 1, 2\}^{H \times W}$. Similarly following [58], we also set $K = 8$ with a training and validation image resolution of $400 \times 400$. The training objective is defined as:

$$\mathcal{L}_{\text{2D}}^{\text{curve}} = \lambda_{CE} \mathcal{L}_{\text{cross-entropy}} + \lambda_{\text{focal}} \mathcal{L}_{\text{focal}} + \lambda_{\text{dice}} \mathcal{L}_{\text{dice}},$$

where $\lambda_{CE}$, $\lambda_{focal}$, and $\lambda_{dice}$ are all set to 1.0, balancing the contributions of cross-entropy, focal [33], and dice [41] losses to the overall training objective.

For the Fusion360 dataset, we utilize 196,050 images from 3,921 shapes. For the DeepCAD dataset, we use 1,745,450 images from 34,909 shapes from the training set. These image datasets are split into training and validation sets at a 9:1 ratio. The Fusion360 dataset's U-Net is trained over 3 epochs, taking 307 minutes, while the DeepCAD dataset's U-Net training for a single epoch requires 20.8 hours, all on a single NVIDIA RTX A6000 GPU.

## E.2  Details of Integrating 2D Segments to 3D Fields

The opacity field $\sigma(\mathbf{x})$ hyper-parameters are set to $\zeta = 10$ and $\beta = 0.8$ as in [68]. For the reconstruction loss of the density field $\mathcal{F}$, we set $\lambda_{sparsity} = 0.5$, $s = 0.5$, $\eta_{\text{batch}} = 0.1$, and $\eta_{\text{image}} = 0.05$. We use a batch size of 8,192 rays for the training of both the density field $\mathcal{F}$ and the semantic field $\mathcal{A}$. We trained all the fields for 1500 iterations on a single NVIDIA RTX 3090 GPU. The whole training time is approximately 5 minutes per shape. Note that the segmentation process

and the learning of object density and semantics are typically done as separate steps, rather than in a single, continuous process.

### E.3 Reverse Engineering Implementation Details

We provide additional details on our reverse engineering process, i.e. from our density and semantic fields to the reconstruction of the extrusion cylinders, as discussed in Sec. 3.4 of the main paper. This process begins by sampling points from the density fields, and then from the extracted point clouds deriving the corresponding parameters for each extrusion cylinder. We query the learned density field value across a $400 \times 400 \times 400$ grid. Points on this grid with an opacity $\sigma$ value greater than a threshold $\tau \geq 0.99$ are selected and their corresponding instance segmentation and start-end(-barrel) segmentation labels are extracted. The extrusion axis ($\mathbf{n}$), height ($\mathbf{h}$), and center ($\mathbf{c}$) are determined directly from the sampled point clouds as detailed in the main paper.

The process to extract the 2D implicit sketch involves additional steps detailed as follows: After predicting the extrusion axis, the 2D sketch boundary points are created by projecting the base curve points along this axis. These 2D point clouds are then used to infer the 2D implicit sketch using IGR [15], which takes as input the projected 2D sketch point clouds and their normals to generate the corresponding signed distance fields. To extract the 2D point normals for IGR, we modify PGR [32] to estimate 2D normals from our projected 2D sketch points. We adapt the algorithm from PGR and modified 2D basis as follows:

$$\tilde{\phi}_j = \begin{cases} -\frac{\mathbf{x}-\mathbf{y}_j}{2\pi|\mathbf{x}-\mathbf{y}_j|^2} & \text{if } |\mathbf{x}-\mathbf{y}_j| \leq w(\mathbf{x}) \\ -\frac{\mathbf{x}-\mathbf{y}_j}{2\pi w(\mathbf{x})^2} & \text{otherwise} \end{cases} \tag{10}$$

where $w(\mathbf{x})$ is the width function defined in PGR. For an in-depth understanding of the formula and its derivations, we direct readers to the PGR paper [32]. Fig. A12 displays the impressive results of our 2D normal estimation implementation, showcasing the accuracy and quality of our approach. These 2D point clouds and the estimated normals are fed into IGR network to output the 2D implicit sketch represented as 2D signed distance function.

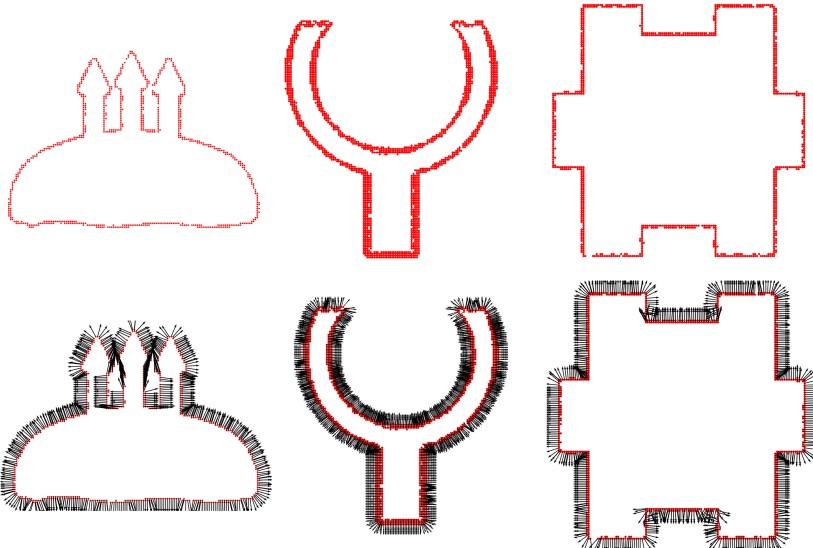

Figure A12: Example of input 2D point cloud (top row) and optimized normal estimation (bottom row).

### E.4 Multi-View Images Preparation Details

We detail the generation of multi-view image datasets used in our experiments. Adapted from [68], the rendering process utilizes BlenderProc [7] and we render 50 views with resolution $400 \times 400$ for each shape, where the viewpoints are sampled by Fibonacci sampling [17] evenly distributed

over a sphere. The surface and curve segmentation labels, we start off by coloring the ground truth mesh black, then marking each segment white. The meshes marked per instance are then rendered in diffuse mode to produce a highlight map for each segment from all viewpoints. Finally, the highlight maps are merged into a single segmentation map, creating multi-view segmentation maps consistently annotated with GT labels across views.

## E.5 NeuS2 [62] + Point2Cyl [58] Implementation Details

We provide additional details on the baseline NeuS2 + Point2Cyl. There are many recent and high-performing studies on converting multi-view images to 3D geometry [40, 61, 44, 13], and we chose NeuS2 because this work is based on InstantNGP [44] and offers fast convergence in reconstructing shapes. For instance, while NeRF [40] takes about 8 hours to reconstruct a sample from the DTU dataset [20], NeuS2 can achieve more accurate geometry in just 5 minutes. It has been confirmed that NeuS2 achieves more accurate geometry than InstantNGP, which has converged in the same duration. We selected this study considering the efficiency and feasibility of evaluation. In the test with NeuS2+Point2Cyl, we used a set of images with 50 view points per shape, at a resolution of $400 \times 400$, and trained for 20,000 iterations. We then ran a marching cube at a resolution of 64 to obtain the mesh, and sampled 8,192 points from that surface to apply as input to Point2Cyl.

## E.6 Binary Operation Prediction

We introduce a straightforward method for recovering binary operations. A naïve approach involves performing an exhaustive search across all possible combinations of primitives and operations, resulting in $2^K$ possibilities for a model with $K$ primitives. To identify the optimal combination that aligns closely with the observed input images, we measure the average per-pixel L2 distance between the rendered images of each combination's resulting model and the observed multi-view images, selecting the combination with the lowest distance. We have implemented this approach to recover the binary operations, and examples of the results are illustrated in Fig. A13.

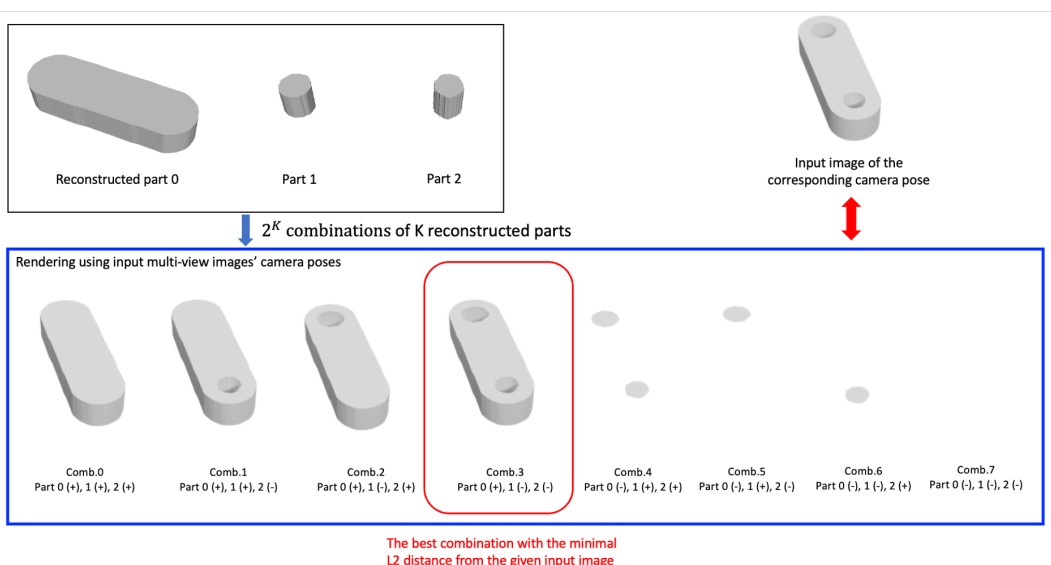

Figure A13: **Example of the binary operation prediction** of the reconstructed extrusion cylinders.

## E.7 Implicit sketch to parametric curves

We also provide example showcases of converting an implicit sketch into a parametric representation. After generating the 2D implicit field for each sketch, we sample boundary points using the marching squares algorithm, which serve as knots for cubic Bézier curves. Using these points, we employ an off-the-shelf spline fitting module from the SciPy library [59] to determine the optimal B-spline representation based on the provided control and knot points. This process is illustrated in Fig. A14.

It is important to note that the resulting sketch may not comprise the minimal set of parametric curves; for example, a single line segment may be represented by multiple segments.

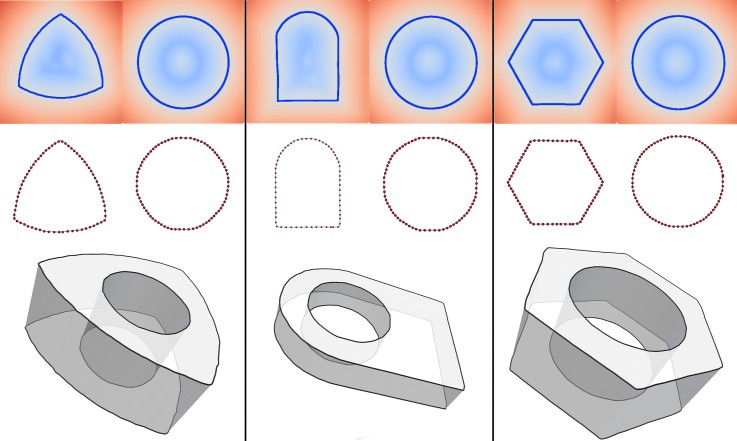

Figure A14: **Recovered parametric models with sketch represented as cubic spline.**

# F    Additional Analysis

## F.1    Comparison of Segmentation Performance Between 2D and 3D Networks

We evaluate the segmentation performance of networks in MV2Cyl and Point2Cyl [58], which process 2D images and 3D point clouds, respectively. The 2D segmentation network in MV2Cyl processes $400 \times 400$ resolution images, categorizing them into background(0), start face(1), end face(2), and barrel face(3). It was tested on 55,400 images from 1,108 shapes. Conversely, the 3D segmentation network in Point2Cyl processes 8,192-point point clouds, segmenting them into barrel(0) and base(1) points, with 1,108 point clouds evaluated. Table A11 details the test accuracy and test accuracy excluding the background for the 2D network. Notable, the 2D segmentation network outperforms the 3D segmentation network, supporting our hypothesis that 2D networks are more effective than 3D networks at segmentation tasks, particularly in extracting the spatial information needed to reconstruct a CAD model.

Table A11: The test accuracy of base-barrel segmentation networks

| Method | Test Accuracy (%) | Test Accuracy Excluding Background (%) |
|---|---|---|
| Point2Cyl | 88.27 | N/A |
| **MV2Cyl (Ours)** | 99.15 | 96.58 |

## F.2    Consistency Between Curve and Surface Segmentations

We compute a consistency metric between the curve and surface predictions. Specifically, we compute the percentage of models where the number of predicted curve instances and surface instances are identical. Across all the test shapes in the Fusion360 dataset, 98% of the models (1,095 shapes over 1,108 shapes) have the same number of curve and surface segments. Moreover, we also measure the consistency between the extracted curve and surface point clouds. Concretely, we measure the average one-way Chamfer distance between the extracted curve point cloud and surface point cloud $(0.081 \times 10^{-3})$. We see that this very small value suggests that the extracted curve point clouds are quite consistent with the surface point clouds.

## F.3    Chamfer Distance Evaluation for 3D Reconstruction

We quantitatively evaluate the 3D reconstruction quality of MV2Cyl by measuring the Chamfer distance against the ground truth CAD model. To generate the output CAD model, binary operations

are determined through a simple exhaustive search approach. The results, shown in Tab. A12, demonstrate that our method outperforms the baseline, NeuS2+Point2Cyl. For a fair comparison, we compare with NeuS2+Point2Cyl instead of NeuS2, as our work focuses on the reverse engineering of extrusion cylinders, which is a more challenging task than 3D reconstruction without CAD structure. Reverse engineering extrusion cylinders enables shape editing and importing the model back into CAD software.

Table A12: **Chamfer distances comparisons using Fusion360 and DeepCAD datasets.** The chamfer distance values are multiplied by $10^3$.

| Dataset | Method | Mean CD ($\downarrow$) | Median CD ($\downarrow$) |
|---------|--------|------------------------|--------------------------|
| Fusion360 | NeuS2+Point2Cyl | 105.18 | 51.88 |
| | **MV2Cyl (Ours)** | 49.08 | 9.40 |
| DeepCAD | NeuS2+Point2Cyl | 106.59 | 17.21 |
| | **MV2Cyl (Ours)** | 18.21 | 1.43 |

## G    Additional Qualitative Results

We provide additional qualitative visualizations of the reconstructed extrusion cylinders using our proposed MV2Cyl. Results are shown in Tab. A13. It showcases MV2Cyl's exceptional ability to process and accurately render shapes of diverse appearances on an instance-wise basis. This demonstration underscores the versatility and precision of our approach in capturing and reconstructing complex geometries from multi-view images.

Table A13: **Visualization of the outputs of MV2Cyl.** It showcases the superior performance of segment the instance and infer the accurate CAD parameters.

| GT Mesh | MV2Cyl | Extrusion Cylinders | | |
|---|---|---|---|---|

| GT Mesh | MV2Cyl | Extrusion Cylinders |
| --- | --- | --- |

| GT Mesh | MV2Cyl | Extrusion Cylinders |
|---------|--------|---------------------|

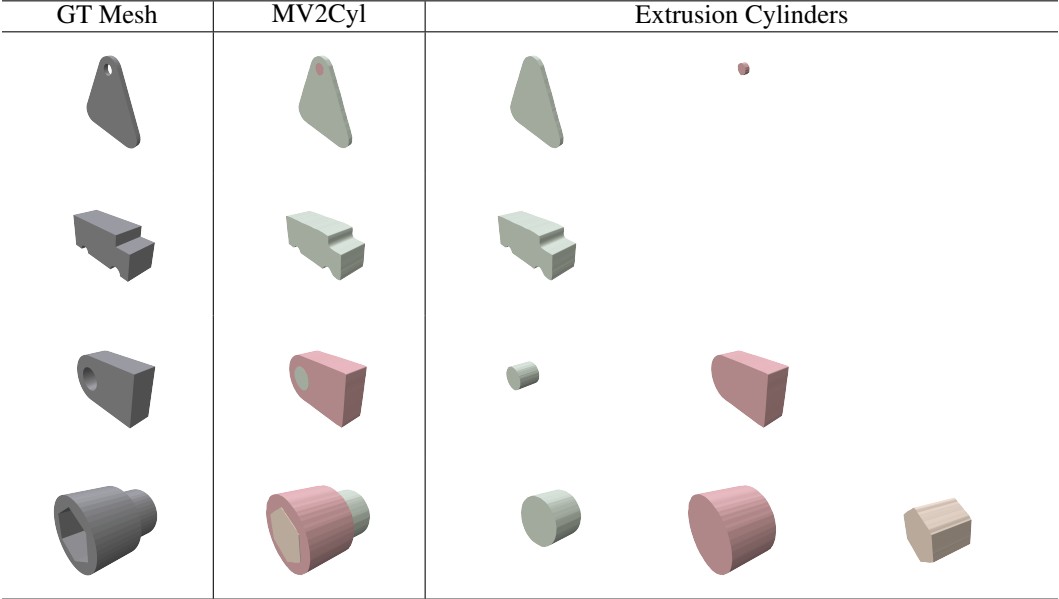

