# OpenReview forum: "MV2Cyl: Reconstructing 3D Extrusion Cylinders from Multi-View Images"
_NeurIPS.cc/2024/Conference — NeurIPS 2024 poster_

### Official Review · Reviewer_dRdb · 2024-07-04

**Soundness:** 3
**Presentation:** 4
**Contribution:** 4
**Rating:** 7
**Confidence:** 5

**Summary:**

The authors introduce a novel method named MV2Cyl, which reconstructs technical 3D  objects withing the sketch-extrude paradigm by utilizing a 2D prior model and a learnable radiance field derived from multi-view images. To accomplish this, the 2D prior model is trained on a labeled dataset to predict semantic labels, $K$ extrusion cylinders, and extrusion curves, which are crucial for further processing of the 3D shape. Once the model is trained, it predicts pseudo-ground-truth for the subsequent stage. In this stage, a radiance field model is trained to replicate the pseudo-ground-truth labels. As the radiance field model converges, it facilitates the extraction of surfaces and curves in 3D space. The authors then detail how to utilize this output to reconstruct the final shape. The experimental results demonstrate that each component significantly contributes to achieving 3D reconstructions that surpass the selected baselines. Given that the radiance field is trained solely from the multi-view images and the prior model, it offers improved real-world applicability compared to previous works.

**Strengths:**

- The paper is overall well-written and easy to follow.
- Each component is effectively justified, and the ablation study outcomes distinctly demonstrate their significance.
- Given the extensive datasets like ABC [1], the prior training represents a significant contribution to this research. It is conceivable that the prior model may serve as a foundational model for CAD extraction in subsequent studies.
- Additionally, the prior and inference models are independent, and both can be adapted adequately if new, better models appear in the literature. This should also facilitate scaling with the datasets.
- I appreciate the demo displayed in the Appendix. It illustrates the practical application of the model and reinforces the societal impact discussed in the final section of the paper.
- One can argue about the pros and cons between different paradigms (such as sketch-extrude, CSG, etc.). The choice of sketch-extrude in this instance is substantiated by the results and outperforms the established baselines.
- Publishing the code and the extended dataset upon acceptance is highly appreciated.

[1] Koch, S, Matveev, A, Jiang, Z, Williams, F, Artemov, A, Burnaev, E, Alexa, M, Zorin, D, Panozzo, DABC: A Big CAD Model Dataset For Geometric Deep Learning. In The IEEE Conference on Computer Vision and Pattern Recognition (CVPR) 2019 .

**Weaknesses:**

- Point2Cyl, ExtrudeNet, and SECAD-Net encode inputs as latent vectors, which are then decoded into the target representation. An encoder in this context could be any model that processes input in a specific modality. Adapting these methods to handle multi-view images appears feasible and would support the paper's claims. Notably, an encoder understanding 3D geometry based solely on 2D images could be advantageous (for instance, utilizing part of the network from [2] as an encoder).
- While the paper is generally understandable, Section 3.4 is hard to follow. Introducing a simple visualization to illustrate the process would aid comprehension.
- In Section 3.4, points 1 and 2 prompt inquiries about the 3D model employed in this study. The authors refer to TensorRF, but it is unclear whether a standard NeRF or NeuS was used. If NeuS was utilized, the predicted normals could be enough to:
    - Determine a fitting plane by voting for the plane that most likely aligns with the normals.
    - Derive the output shape using a standard algorithm like ball-pivoting or Poisson surface reconstruction, both of which are widely accessible.
- Furthermore, a NeuS backbone would arguably be more appropriate for the application described in the paper, as it more precisely reconstructs surface fields, potentially enhancing the accuracy of the curve field.

[2] Watson, D, Chan, W, Martin-Brualla, R, Ho, J, Tagliasacchi, A, Norouzi, M. "Novel view synthesis with diffusion models". arXiv preprint arXiv:2210.04628 2022.

**Questions:**

Questions:
- I will refer to:
> We first extract the surface point cloud and curve point cloud from the corresponding 3D fields by thresholding the corresponding existence fields. Each point in the point cloud would have attributes queried from the corresponding attribute fields.

How is the point cloud sampled specifically? Is it accurate the underlying radiance field uses density instead of SDF field like in NeuS?

Suggestions:
- The explanation in lines 44-47 assumes prior knowledge of the sketch-extrude principle. Without this knowledge, it's challenging to understand the source of the issue. An inset visualization could greatly aid comprehension.
- I propose renaming the "existence field" and "attribute fields". The term "existence field" implies a binary output indicating whether an object exists at a given point. "Density field" would be more appropriate, clarifying how prior values translate to opacity and aligning with the field's intended purpose.
Regarding the "attribute field", the term could be misconstrued as referring to "attributes" such as colors. "Semantic field" would be more accurate.
- It can be deducted from the context that $L_\text{existence}$ applies to both curve and attribute fields. However, stating that clearly in the text would remove potential ambiguities for other readers.

**Limitations:**

The potential negative societal impact is clearly stated and supported by the real-demo presented in the Appendix.

Regarding the limitations, it is unclear what authors mean by:
> our framework is limited in predicting binary operations across primitives. We plan to further explore predicting binary operations using multi-view image inputs.

in lines 328-331. Some examples or an visualization in the Appendix would be appreciated here. It would be worth adding a comment on the shape complexity the model cannot handle.

---

> ### Author Rebuttal · Authors · 2024-08-07
>
> - **Adapting methods to multi-view images (Image encoder + CAD decoder).**
> Thank you for the suggestion. First, we want to note that for Point2Cyl and SECAD-Net, it is not feasible to directly replace the point cloud encoder with an image encoder since the encoder does not simply output a latent code but assigns information to each point, such as the segment label or surface normal. ExtrudeNet is the only case that produces a latent code from the input shape. While we couldn't try replacing the point cloud encoder of ExtrudeNet with an image encoder during the tight rebuttal period (as we needed to address questions from five reviewers), we promise to include the results of this experiment in the revision. We strongly believe that our method—segmenting 2D images, unprojecting them to 3D space, and fitting extruded cylinders rather than directly predicting them—will demonstrate superior performance.
> - **Section 3.4. Simple visualization.**
> Thank you for your suggestion. We add a simple visualization to clarify the pipeline of the reverse engineering of CAD parameters as shown in Figure 1.
> - **Clarification on 3D model (TensorRF) employed.**
> Our approach directly builds on top of TensorRF which is based on standard NeRF that models a density field directly instead of an SDF as in NeuS. We will clarify this in the final version. We find that this choice of using TensorRF works well in our setting and is also fast to optimize. We thank the reviewer for their suggestion on using a NeuS backbone, which is interesting and also makes sense. We leave this exploration for future work.
> - **How are point clouds sampled from the learned fields?**
> We refer to Appendix A5.3 (Ln 638-642). We query the learned existence field by sampling points across a 400x400x400 grid. The existence values  $\mathcal{F} (x)$ are converted into opacity values (Eq. 3), and we keep the sampled points whose opacities are greater than 0.99, obtaining our point cloud.
> - **Adding visual aids for lines 44-47.**
> Thank you for the suggestion. We will include an additional visualization to aid comprehension in the final version.
> - **Renaming “Existence field” and “Attribute field” to “Density field” and “Semantic field”.**
> Thank you for your suggestion. That makes sense! We will modify this in the final version.
> - **Clearer statement of L_existence.**
> Thank you for the suggestion. We will state this explicitly and more clearly in the text of the final version.
> - **Clarification on limitation (line 328-331)**
> Similar to Point2Cyl, our approach did not explicitly predict the binary operations of the primitives, i.e., determining whether each primitive is 'added' or 'subtracted' from the model. We will make this statement more clear in the final version. Note that in this rebuttal, we further provided a simple approach to the binary operation recovery: one naive and simple approach is to do an exhaustive search against all possible primitives-operations combinations (2^K possibilities for a model with K primitives), and take the best combination, which is closest to the observed (input) images. To obtain the best combination, we measure the average L2 distance between the rendered images of each combination’s resulting model against the observed multiview images and select the lowest one. We implement this approach and obtain the binary operations, examples are shown in Figure 3.
> - **Failure cases.**
> Since MV2Cyl relies on 2D image inputs, it’s susceptible to occlusion. Specifically, if one side of an extrusion cylinder is totally hidden by the other ones, our 2D segmentation model cannot catch the hidden side. In this case, the extrusion cylinder cannot be reconstructed. In Figure. 5, the left shape is the target CAD model and the right one is the reconstructed CAD model by MV2Cyl. While the target shape has an inset hexagonal cylinder whose one end is hidden by the outer one, MV2Cyl failed to reconstruct the inset extrusion cylinder.

---

> > ### Comment · Reviewer_dRdb · 2024-08-09
> > **RE: Rebuttal**
> >
> > I thank the authors for their response and for their effort to provide additional experiments. The attached PDF rebuttal promises a significant changes to the final version and I'll be more than happy to see that paper accepted to the conference.

---

> > > ### Author Response · Authors · 2024-08-14
> > >
> > > Dear Reviewer dRdb,
> > >
> > > We're glad our rebuttal addressed your questions. We appreciate your detailed feedback, which greatly helped improve our work.

---

### Official Review · Reviewer_7GUw · 2024-07-11

**Soundness:** 3
**Presentation:** 3
**Contribution:** 2
**Rating:** 5
**Confidence:** 3

**Summary:**

This paper proposes a new method to reconstruct extrusion cylinders from multiview images. The key idea is to train a CNN to predict the binary mask for each surface and the sketch of each surface. Then, these predictions are used in learning 3D neural fields for each surface and sketches by the volume rendering technique.

**Strengths:**

1. The task seems to be interesting in that we use multiview images with semantic segmentation to help reconstruct 3D extrusion cylinders. This method is novel for me.
2. The experiments demonstrate some improvements over previous reconstruction methods.

**Weaknesses:**

1. How to associate each surface is unclear here. If I understand it correctly, the predicted labels are similar to instance segmentation labels so we may have multiple different labels for the same region. How to associate predicted surfaces with different viewpoints is not clear here.
2. The prediction of feature lines could be sensitive to the line width. If we use a large line width the localization of the feature lines could be difficult because we cannot learn a perfect 3D field for these lines. If we use a small line width, learning a feature field for signals just on very sparse pixels could be problematic and we may extract disconnected feature lines (sketches) here. The setting of the linewidth for training could be tricky here.
3. It is difficult to get some real-world images to train these CNNs, which could harm the generalization ability of the proposed method. If we have painted some textures on these CAD models, then the proposed method could fail because it is not trained on models with textures.
4. The proposed method seems to be too concentrated on a specific problem in computer graphics. Predicting extrusion cylinders is not a common task in daily life and the proposed method does not contain technically new ideas on neural networks. The paper seems to be more suitable for some graphics journals or conferences rather than NeurIPS.

**Questions:**

Refer to Weakness.

**Limitations:**

Limitations are discussed in the paper, which is the binary prediction of CNNs. I think the method could be sensitive to hyperparameters, which could be a potential limitation.

---

> ### Author Rebuttal · Authors · 2024-08-07
>
> - **Surface label association.**
> Yes, you are right that the predicted segmentation maps of the multi-view images are instance segmentation labels. Hence we use Hungarian matching (see Ln 209-215 main paper) to align the labels between the segmentation maps of the training images with the segmentation labels of the attribute field at training time.
> - **Line width sensitivity.**
> We use a line width of 5 pixels in our experiments, which we found works quite well in our setting. We further experiment on using different line widths, specifically 2.5 pixels (50% thinner) and 7.5 pixels (50% thicker). Results are shown in Table 4, where we see that our approach is not that sensitive to line width.
> - **Textured CAD models.**
> Yes, our current method does not directly handle textured CAD models. We plan to explore handling textured objects by utilizing generalizable large 2D models such as Segment-Anything (SAM) to extract the object mask removing the texture and similar to our current real data experiments we can remove the background, which together can potentially bridge the domain gap. We leave this exploration as future work.
> - **Specific problem with computer graphics.**
> In this work, our main idea is to demonstrate how 2D priors can aid in reconstructing 3D structures when combined with neural rendering and to explore the best way to integrate different 2D priors (such as surfaces and curves) in 3D space. Although we focus on reconstructing extrusion cylinders in this study, we believe that the ideas presented have the potential to influence future research in many areas of machine learning. We will revise the introduction of our submission to clarify this point.
> - **Limitations: binary operation prediction.**
> One simple approach to predicting binary operations without CNNs is to do an exhaustive search against all possible primitives-operations combinations (2^K possibilities for a model with K primitives) and take the best combination, which is closest to the observed (input) images. To obtain the best combination, we measure the average per-pixel L2 distance between the rendered images of each combination’s resulting model against the observed multiview images and select the lowest one. We implement this approach and obtain the binary operations, examples are shown in Figure 3. We leave further explorations for better approaches, such as being cheaper and faster, as future work.

---

### Official Review · Reviewer_mRwh · 2024-07-11

**Soundness:** 3
**Presentation:** 3
**Contribution:** 2
**Rating:** 5
**Confidence:** 4

**Summary:**

MV2Cyl is a method that proposes to solve the 3D reverse engineering of CAD models. The network takes as input multi-view images and outputs extrusion cylinders. The method extends Point2Cyl [58] that proposed to predict extrusion cylinder from point clouds. It is argued that multi-view images can easily be obtained from 3D scans and that 2D CNNs have superior performance than 3D processing networks.
The network is composed of three main components. Firstly, 2D CNN U-Net-based networks learns to segment and classify the pixel image of the 2D images into surface and curve segments. Then, a NERF based approach is used to leverage the 2D segments into 3D existence and attribute fileds. Finally, 3D surface and curve point clouds are extracted from the 3D fields. The extrusion parameters (extrusion axis, height and centroid) and the sketch (implicit function) are estimated from the extracted point clouds.
The method is evaluated on the DeepCAD[64] and Fusion360[63] datasets, from which multi-view images are extracted. The results show an improvement compared to Point2Cyl[58] and a Neus[51]+Point2Cyl.

**Strengths:**

- The paper is fairly clear and easy to understand for a reader that is familiar with Point2Cyl [58].

- The main strength of the proposed work is that it shows an improvement to the results presented in Point2Cyl [58]. This is achieved by leveraging the information provided in multi-view images as opposed to 3D point clouds.

**Weaknesses:**

- While the proposed work offers improvements to Point2Cyl [58], the reviewer feels that it does not address the main limitations of Point2Cyl. Firstly, the binary operations between cylinder (how to assemble the cylinders into the GT CAD model) are not predicted. This limitation is mentioned in the paper but addressing it would have lead to a significant improvement to Point2Cyl. Secondly, the sketches are predicted as implict functions and not parametric curve primitives such as in real CAD models. This produces noisy surfaces (as shown in Figure 3 of the paper), which are quite different from the standard sharp representation of CAD models.

- The evaluation consists of a comparison with Point2Cyl using some of the metrics proposed in Point2Cyl. Nevertheless, further analysis of the results could be completed. For example, it could be meaningful to compute a consistency metric between the surface and curve predictions. One would expect the two network to predict the same number of extrusion cylinders. Also, are the extracted curve point clouds consistent with the surface point clouds? A comparison to the point cloud to CAD sequence experiment presented in DeepCAD[64] could also be included as the extrusion cylinders can easily be extracted from the CAD sequences. It would also be interesting to know how the performance of the network changes with respect to the number of extrusion cylinders (K) present in the ground truth.

**Questions:**

- The representation of the output is a little unclear. Figure 3 suggests that the output of MV2Cyl is a CAD model. Can a CAD B-Rep be obtained from the output? If so, how are the sketch primitive and parameters obtained from the predicted implicit functions?
Beyond the points mentioned in the weaknesses, it would be interesting to know how the perfomance of the model changes with respect to the number of input images at test time.

- Many qualitative results are presented in the Appendix, all of them show very impressive predictions. It would also be interesting for the reader to see examples for which the model fails to recover the correct shape.

**Limitations:**

- The authors have addressed one of the main limitations of the work that is the lack of prediction of the binary operations between the different cylinders. Potential societal impacts have also been included.

---

> ### Author Rebuttal · Authors · 2024-08-07
>
> - **Binary operations.**
> The binary operations can be recovered in a simple/straightforward approach such as an exhaustive search against all possible primitives-operations combinations (2^K possibilities for a model with K primitives). We take the best combination as the output configuration, which is the combination that is closest to the observed images. To obtain the closest/best combination, we measure the average per-pixel L2 distance between the rendered images of each combination’s resulting model against the observed multiview images and select the lowest one. We implement this approach and obtain the binary operations, examples are shown in Figure 3. We will include this in our final version.
> - **Sketches: from implicits to parametric curves.**
> We demonstrate how to convert an implicit sketch to a parametric representation. After acquiring the 2D implicit field for each sketch, we sample boundary points using the marching squares algorithm and use them as knots for cubic Bézier curves. Utilizing these points, we employ an off-the-shelf spline fitting module implemented in Scipy library which finds the best B-spline representation with the given control/knot points. This process is illustrated in Figure 4. Note that the resulting sketch may not consist of the minimal set of parametric curves; for example, a single line segment might be divided into multiple segments. Optimizing this aspect is left for future work.
> - **Analysis of results: consistency metric.**
> Thank you for your suggestion. We compute a consistency metric between the curve and surface predictions. Specifically, we compute the percentage of models where the number of predicted curve instances and surface instances are identical. Across all the test shapes in the Fusion360 dataset, 98% of the models (1,095 shapes over 1,108 shapes) have the same number of curve and surface segments. Moreover, we also measure the consistency between the extracted curve and surface point clouds. Concretely, we measure the average one-way chamfer distance between the extracted curve point cloud and surface point cloud (0.081 x 10^-3). We see that this very small value suggests that the extracted curve point clouds are quite consistent with the surface point clouds.
> - **Comparison against DeepCAD’s point cloud to CAD.**
> Thank you for suggesting a comparison with DeepCAD’s point-cloud-to-CAD framework. We attempted this comparison but encountered several challenges. First, the pretrained model for DeepCAD’s point-cloud-to-CAD framework was not available. Second, retraining the DeepCAD model with our dataset took several days, which was difficult to manage within the short rebuttal period. Third, we tried using an unofficial pretrained model from GitHub, but the resulting CAD outputs were completely different from the input point clouds, suggesting possible bugs in the training process. We plan to contact the authors of DeepCAD to resolve these issues and achieve a fair comparison.
> Despite these challenges, we strongly believe that our method—segmenting 2D images, unprojecting them to 3D space, and fitting extruded cylinders rather than directly predicting them—will demonstrate superior performance.
> - **Performance w.r.t. different number of extrusion cylinders (K).**
> Table 1 shows the results breakdown of the models in the Fusion360 test set across different numbers of extrusion instances (K). We see that the general trend is that the error steadily increases as with larger K, i.e. as model difficulty increases. (Note that the number of samples for K=7, 8 are very small; please take these statistics sparingly.)
> - **Output representation.**
> The output representation is a set of extrusion cylinders. Specifically, for each extrusion cylinder, we have its center (R^3), extrusion axis (SO(3)), extent (R^2) and a sketch represented as an implicit (which can also be converted into a set of parametric curves (see above response). Furthermore, for each extrusion cylinder, we can also recover its operation, which is either additive or subtractive, (see “Binary operations” response).
> - **Ablation on the number of input images.**
> Table 3 shows an ablation of MV2Cyl for a different number of input images at test time. Results show that our approach is not that sensitive to the change in number of input images and still achieves reasonable performance for as low as 10 input views.
> - **Failure cases.**
> Since MV2Cyl relies on 2D image inputs, it’s susceptible to occlusion. Specifically, if one side of an extrusion cylinder is totally hidden by the other ones, our 2D segmentation model cannot catch the hidden side. In this case, the extrusion cylinder cannot be reconstructed. In Figure. 5, the left shape is the target CAD model and the right one is the reconstructed CAD model by MV2Cyl. While the target shape has an inset hexagonal cylinder whose one end is hidden by the outer one, MV2Cyl failed to reconstruct the inset extrusion cylinder.

---

> > ### Comment · Reviewer_mRwh · 2024-08-13
> > **Answer to author rebuttal**
> >
> > The reviewer thanks the authors for the different clarifications and additional results. Most of the concerns were properly addressed by the authors. As for the DeepCAD pretrained point-to-CAD model, the authors could have used the pretrained model (https://github.com/ChrisWu1997/DeepCAD?tab=readme-ov-file#pre-trained-models) and trained their own multi-view image encoder to learn the latent space of the pretrained CAD sequence autoencoder. Based on the elements of the rebuttal, the reviewer raises the rating from 4 to 5.

---

> > > ### Author Response · Authors · 2024-08-14
> > >
> > > Dear Reviewer mRwh,
> > >
> > > We are pleased that our rebuttal has clarified the points you raised, and we greatly appreciate the insights you provided.
> > >
> > > Regarding the link to the pre-trained models, we found that it does not include the point cloud encoder. We were unable to train the point cloud and multi-view image encoders during the short rebuttal period while allocating our time and computational resources to handle numerous requests for additional experiments from five reviewers. However, we strongly believe that our method, based on surface and curve prediction, will outperform the simple autoencoding architecture. We promise to include this additional experiment in the final version.

---

### Official Review · Reviewer_AtoY · 2024-07-11

**Soundness:** 3
**Presentation:** 4
**Contribution:** 2
**Rating:** 5
**Confidence:** 4

**Summary:**

This paper proposes a method to predict extrusion cylinders from images of a CAD part. Specifically, it takes a set of masked multi-view images as input; these are then processed independently by instance segmentors trained to find extrusion curves and surfaces. Neural fields are then fitted, that reconstruct these 2D segmentations in terms of 3D segmentations. Lastly, a set of primitives explaining the 3D segmentations is extracted using a series of heuristics (RANSAC and other hand-crafted robust estimators). Experiments are conducted on two standard benchmark datasets (Fusion 360 and Deep CAD), and show the proposed method out-performing a baseline approach (NeuS reconstruction followed by an existing 3D primitive-prediction pipeline).

**Strengths:**

- The work presents a sophisticated and carefully engineered pipeline, with choice of components clearly motivated.
- The approach of fitting several 3D fields to the 2D segmentations, and regularising to ensure these are binary, is interesting.
- There is a fairly extensive evaluation (qualitative and quantitative) on synthetic (rendered data), including comparison against a baseline that first performs 'naive' surface reconstruction, then applies a 3D-only primitive extraction to find the extrusion cylinders.
- Quantitatively, the proposed method is found to significantly out-perform the reconstruction-then-fitting baseline. Indeed, it also out-performs a method that directly predicts extrusion cylinders from ground-truth 3D point-clouds, which is a notable success since the latter would seem to have access to more complete information
- Qualitatively, the proposed method appears to correctly capture geometric features of the input objects, using an appropriate (minimal) set of primitives, and providing much cleaner reconstructions than a naive
- The paper is well-written, sensibly organized and pleasant to read throughout.

**Weaknesses:**

- The work focuses on clean synthetic (rendered) inputs, with only one experiment (in the appendix) on real data. However, operating on multi-view images offers no benefit over 3D-input methods when one has access to the 3D shapes (e.g. as point-clouds). For a strongly 'applied' paper like this, the lack of focus on the practically-relevant scenario (i.e. actual photos) is problematic.
- Real-world results (though still with in-distribution CAD shapes 3D-printed from the datasets) are not very impressive, despite fine-tuning for this setting. There is also no quantitative evaluation her, which is a pity since this would be straightforward given that the ground-truth CAD models are available (modulo 3D printing imperfections)
- There is no quantitative evaluation of the quality of 3D reconstruction in the sense of distance (e.g. hausdorff / f1) from the reconstructed shape to the original CAD model. It would be valuable to add this (and compare with NeuS2), to give an idea of how well the shapes are being predicted in a more universally-meaningful metric, as opposed to the given metrics that are specific to the shape representation and difficult to interpret in a broader context.
- There are few generalisable insights present in the paper. It presents a pipeline that evidently works well for the specific task, but the technical contributions are very narrow in applicability, reducing potential impact.
- Overall the paper is a poor fit for NeurIPS – lots of engineering (obviously an achievement in itself), but minimal machine learning (just training a segmentation model on synthetic data, then fitting the single-scene neural fields), making the interest to the community relatively low.

**Questions:**

How/where do you predict whether each primitive is 'added' or 'subtracted' from the model? The results in Fig. 3 indicate that the model correctly inserts holes, but it is unclear in the method where this determination is made

**Limitations:**

These are discussed briefly but adequately in Sec. 5.

---

> ### Author Rebuttal · Authors · 2024-08-07
>
> - **Work focuses on clean synthetic data; lacking real data experiments.**
> We note that it is difficult to obtain real data with ground truth CAD (sketch-extrude) parameters. In fact, to the best of our knowledge, such a dataset does not currently exist. Hence we opt to use synthetic data to train and test our method. We provide more real data examples in Table 6 (rebuttal) which supplements the real-world examples in Table A3 (supp).
> - **Benefits of multi-view images over 3D inputs.**
> We note that images are easier to capture and acquire such as with our mobile phones or digital cameras compared to 3D point clouds that require a depth camera or LiDAR scanner. Moreover, we also compared directly with Point2Cyl, which takes 3D point clouds as input, in Table 1 (main) and showed that MV2Cyl achieves better performance due to the superior performance of 2D networks compared to 3D backbones on the segmentation task, as shown in Table A7 (suppl.).
> - **Quantitative results on real data.**
> We quantitatively evaluate MV2Cyl on real data found in Table A3 (supp). Our reconstruction achieves an average Chamfer distance of 2.11*10^-3 from the ground truth CAD model. To compute this distance, we first needed to align the reconstructed model to the corresponding ground truth shape with the reconstructed model using ICP, after which the desired metrics can be computed.
> - **Quantitative evaluation on 3D reconstruction using distance-based metric.**
> We quantitatively evaluate the 3D reconstruction quality of MV2Cyl by measuring the Chamfer distance against the ground truth CAD model. To obtain the output CAD model, we obtain the binary operations through a simple exhaustive search approach (see “how to predict binary operations” below). Results are shown in Table 5, where our approach shows superior performance against the baseline, NeuS2+Point2Cyl.  We note that for fair comparison, we compare with NeuS2+Point2Cyl instead of NeuS2 as the focus of our work is the reverse engineering of extrusion cylinders, which is a harder task compared to just 3D reconstruction without the CAD structure. Reverse engineering of extrusion cylinders allows for shape editing as well as importing the model back into CAD softwares.
> - **Generalization insights to the paper/narrow applicability.**
> Thank you for acknowledging the generalizable insights presented in our paper. We believe that our approach, which leverages 2D image segmentation and neural rendering for 3D structure reconstruction, can be extended to a broader range of CAD models and thus inspire future research. Additionally, extrusion cylinders are a representation widely used in the CAD industry, and their reconstruction has also been extensively studied, as described in our related work section.
> - **Fit to Neurips.**
> In this work, our main idea is to demonstrate how 2D priors can aid in reconstructing 3D structures when combined with neural rendering and to explore the best way to integrate different 2D priors (such as surfaces and curves) in 3D space. Although we focus on reconstructing extrusion cylinders in this study, we believe that the ideas presented have the potential to influence future research in many areas of machine learning. We will revise the introduction of our submission to clarify this point.
> - **Binary operation prediction: determining whether each primitive is 'added' or 'subtracted' from the model.**
> Similar to Point2Cyl, our approach did not explicitly predict the binary operations of the primitives. We further provide a simple approach to the binary operation recovery: one naive approach is to do an exhaustive search against all possible primitives-operations combinations (2^K possibilities for a model with K primitives) and take the best combination, which is closest to the observed (input) images. To obtain the best combination, we measure the average per-pixel L2 distance between the rendered images of each combination’s resulting model against the observed multiview images and select the lowest one. We implement this approach and obtain the binary operations, examples are shown in Figure 3. We will include this in our final version.

---

> > ### Comment · Reviewer_AtoY · 2024-08-12
> > **Response to rebuttal**
> >
> > I thank the authors for their detailed rebuttal. The quantitative 3D metrics in particular are valuable and show a benefit to the proposed method. The additional results on real images are also appreciated, though I still find the results disappointing (similar to those already provided in the main paper) -- perhaps inevitable due to the reliance on synthetic data for training. As such I'm not going to increase my rating, but I'm still marginally in favor of acceptance.

---

> > > ### Author Response · Authors · 2024-08-14
> > >
> > > Dear Reviewer AtoY,
> > >
> > > We're pleased our rebuttal addressed your questions and highlighted the benefits of our method. We greatly appreciate your detailed feedback, which has significantly improved our work.
> > >
> > > Regarding real data results, techniques for bridging domain gaps or using more realistic textures and backgrounds in training data could improve the quality. We plan to explore these approaches in future research. Thank you for recognizing the potential and value of our work in proposing a novel method for multi-view 3D CAD reconstruction.

---

### Official Review · Reviewer_Ajxc · 2024-07-13

**Soundness:** 3
**Presentation:** 4
**Contribution:** 3
**Rating:** 5
**Confidence:** 4

**Summary:**

## Summary of the Paper:

*Problem Statement*:

     Given multi-view images (RGB) of a 3D shape, the paper aims at recovering a *set* of extrusion cylinders to represent the underlying 3D shape.


*Motivation*:

     Existing works that address 3D shape reconstruction problem through sketch-extrude take raw 3D geometry as input (ex: Point2Cyl) whereas this work takes multi-view images. Other works that take multi-view images as input perform parametric reconstruction (ex: NEF), which while important, does not reconstruct the 3D surface itself. And so, there exists a gap in the community in terms of either the input representation, or the output result, in the context of 3D reconstruction through sketch-extrude modeling. This forms the basis of this work, termed, MV2Cyl.

*Contributions*:

     Developing a method for sketch-extrude reconstruction of 3D shapes from multi-view images, via labeled surfaces and curves.


*Input*:

     Multi-view RGB images of a man-made 3D shape

*Output*:

     3D reconstruction of the object based on sketch-extrude cylinders. A cylinder is

*Dataset used*:

     Fusion 360 and DeepCAD

*Underlying Modeling Tool*:

1)	 U-Net for processing images (segmenting curves and surfaces)

2)	 Implicit field-based volumetric rendering for integrating 2D estimated curve and surface information to corresponding 3D space


*Learning Mechanism*:

     Strongly supervised


*Loss Functions*:

     For 2D surface segmentation, a U-Net with multi-class cross-entropy loss constrained by Hungarian matching algorithm is used. The Hungarian matching constraint is added to account for order invariance of the extrusion segments on the rendered 3D shape, as well as the to classify start-end-barrel segments.

     The 2D curve segmentation module also follows the same architecture and loss functions. In addition, due to the sparsity of labels for curves (which can cause the network to just “attending” to the background pixels), two additional loss terms are used – Dice Loss and Focal Loss

     2D to 3D integration using volumetric rendering: Here, the paper describes two fields. One is the Existence field and the other is the Attribute Field.

     The training of the Existence field is dictated by the input 2D images, which is accomplished via differentiable rendering. A weighted combination of L2 loss (for surface) and sparsity loss (for curves) is used.

     The training of the Attribute filed is governed by a multi-class cross-entropy loss function, constrained by Hungarian matching.


*Quantitative Metric*:

     Extrusion-axis error (E.A.), extrusion-center error (E.C.), per-extrusion cylinder fitting loss (Fit Cyl.), global fitting loss (Fit Glob.)


*Baselines*:

     Since existing works do not take multi-view images for sketch-extrude 3D reconstruction tasks, no suitable comparison exists. However, the paper compares against a closely related work (and its variant), Point2Cyl (+NeuS2), which takes a point cloud as input.

**Strengths:**

1)	 Well-written paper

2)	 Suitable comparisons have been made with Point2Cyl (and +Neus2) to investigate the apparent gains rendered by the proposed framework

3)	 Experimental results on two relevant and complex CAD datasets (Fusion-360 and DeepCAD) are better than other closely related works

**Weaknesses:**

1)	 I wanted to understand how essential the 2D curve segmentation network is in making the reverse engineering possible. In other words, how would the cylinder extrusion results in 3D look like if this module were not present? This is a critical piece of experiment, something that should have been shown through ablation experiments. A similar ablation experiment for 2D surface estimation is NOT necessary though, since it is straight-forward to see why this is needed in the first place, given multi-view images as inputs.
2)	 There is a lack of discussion on which kind of shapes does the proposed approach fail on. I would have liked to see a figure (with MV images, 3D GT and sketch-extrude 3D reconstructions) that shows the kind of shapes that MV2Cyl fails at, even if such results are just a tiny sample. This is consequential to the research push in this direction, and also does not do complete justice to the paper.
3)	 In Table 1, the different quantitative metrics lack corresponding units. For ex., as I understand, the extrusion axis error (E.A) is an angle measurement. The reported number must have been in degrees, yes? Likewise, it would be good to see units of measurement, if applicable, for other metrics reported therein. If you want to save space, you could just put it in the table, instead of mentioning in the caption.
4)	 The extrusion instance segmentation is performed by both 2D segmentation modules (Surface and Curve). I sense that there is a strong dependency on this task to help complete the other “auxillary” task in these modules. As this (extrusion instance segmentation) is a common task for the two, were there experiments done to assess if these two modules (Surface and Curve) could be merged? To me, there is something amiss. Would like to see an intuitive explanation justifying this two-step, common-task-occurrence justification.

**Questions:**

1)	 I do not see RGB inputs (as claimed on L 105) in the Teaser as well as Fig 1. I see just a grayscale image. Which one is it?
2)	 How do you differentiate NEF (CVPR 2023) from the 2D curve segmentation framework? The curves that are of interest can be weakly considered as edges. Can you replace 2D curve segmentation network with NEF (part/whole) to detect the extrusion curves and the base curves?
3)	 Dataset question (L259- 261)– how were these datasets enriched with 2D segmentation maps? That is, how were the ground truth 2D curve and extrusion segments obtained? Manually? Provide details.
4)	 I am curious to know why ABC dataset was not considered. It is a highly relevant dataset that would have made the comparisons exhaustive. What was the reasoning here?
5)	 Can you talk about the extendibility of this approach? I ask this question since I find it non-trivial to train the Existence and Attribute fields, as described in the paper.

**Limitations:**

Yes, the authors have correctly addressed the limitations of the work, including any societal impact that may arise as a result of the publication.

---

> ### Author Rebuttal · Authors · 2024-08-07
>
> - **2D curve segmentation network ablation.**
> See Table A2 and Figure A4 in the supplementary for an ablation on the necessity of the 2D curve segmentation network; referred to as the “Surface only” approach. We see that without the curve segmentation module, occlusions between the extrusion cylinder instances can result in missing base faces leading to poor reconstruction.
> - **Failure cases.**
> Since MV2Cyl relies on 2D image inputs, it’s susceptible to occlusion. Specifically, if one side of an extrusion cylinder is totally hidden by the other ones, our 2D segmentation model cannot catch the hidden side. In this case, the extrusion cylinder cannot be reconstructed. In Figure. 5, the left shape is the target CAD model and the right one is the reconstructed CAD model by MV2Cyl. While the target shape has an inset hexagonal cylinder whose one end is hidden by the outer one, MV2Cyl failed to reconstruct the inset extrusion cylinder.
> - **Units for quantitative metrics.**
> Yes, the reported extrusion axis error is in degrees. We will include the corresponding units in the final version.
> - **Merging the 2D segmentation modules.**
> Thank you for the suggestion. We experiment with combining the 2D curve and surface segmentation networks. We use a shared U-Net backbone that then outputs two separate branches for curve and surface segmentation. Results on using merge and separate UNets to extract the 2D segmentation maps for our reverse engineering extrusion cylinders task are shown in Table 2. While we see that there is a small improvement in performance on some of the metrics (Fit Cyl. and Fit Glob.), the results on both approaches are comparable. We will include these results in the final version.
> - **Clarification on RGB input.**
> Our segmentation networks take as input a 3-channel image, i.e. RGB. Our training data was constructed by rendering **untextured** models using Blender, which is why the images look like they are grayscale. For real data that may have texture, we first convert the input image into grayscale to bridge the domain gap. We find that this does not harm the performance and we are still able to achieve reasonable curve and surface segmentations as shown in Figure 2.
> - **Difference between our curve module and NEF.**
> The main difference between our curve module and NEF is NEF learns a view-dependent color term alongside its edge density field, while our curve module learns a semantic/curve instance segmentation field together with the curve/edge density field. We further note that for the 2D network, NEF utilizes PiDiNet as an edge detection network, while our approach trains a U-Net for curve instance segmentation.
> - **How the 2D curve segments are obtained.**
> See Section A5.4 in the supplementary materials for details on multi-view image and curve/surface segmentation map data preparation.
> - **ABC dataset.**
> We ran our method on the DeepCAD dataset, which is a **subset of the ABC dataset** as provided by [Wu et al., 2021] that only includes the sketch-extrude models.
> - **Extendibility of the approach; non-trivial to train existence and attribute fields.**
> Thank you for the inquiry. Could you clarify what you mean by extendability? We are happy to answer this in the discussion phase. On training the existence and attribute fields, given obtained 2D curve and surface segmentation maps, e.g. from our U-Net, the optimization of the existence and attribute fields becomes straightforward, that is a standard NeRF-model optimization.

---

> > ### Comment · Reviewer_Ajxc · 2024-08-14
> >
> > Authors,
> >
> > Appreciate the attempt to answer all of my questions.
> >
> > Re. Extendability, the term is self explanatory. But for your clarity, I will elaborate -- By extendability, I was referring to how easily this approach can modified/"rewired"? As well, how easily can this be plugged into another design/framework to tackle a similar problem?
> >
> > I am aware that the Existence and Attribute fields undergo standard NeRF-model optimization. But do you see any "handling" and "portability" issues as you have two NeRF models being used for the task? All of these questions fall under the hood of "extendability". Let me hear your thoughts.
> >
> > I would stick to my original rating of BA still.

---

> > > ### Author Response · Authors · 2024-08-14
> > >
> > > Dear Reviewer Ajxc,
> > >
> > > Thank you for your valuable feedback and for helping to improve our work.
> > >
> > > We appreciate your elaboration on "extendibility" in this context. MV2Cyl could indeed be extended to other tasks requiring integrated 2D-to-3D reconstruction, such as B-Rep modeling from multi-view images. Specifically, our 2D segmentation model can decompose curve/surface patches into segments for B-spline fitting, utilizing datasets like ParSeNet (ECCV 2020). The volume rendering module can then reconstruct 3D points, which can be fitted into B-splines using tools like the NURBS-Python library. Thank you for the feedback, and we will further explore this direction.
> > >
> > > Regarding the use of separate volume-rendering models, we did not observe any specific issues with handling the two NeRF models or porting between them. If your concern is about the consistency between the two NeRF models, we kindly ask you to refer to our rebuttal to Reviewer mRwh, where we present the consistency metrics between the curve and surface reconstruction models. If you have any other specific concerns about handling and portability, please elaborate, and we will be happy to address them.

---

### Author Rebuttal · Authors · 2024-08-07

We appreciate the invaluable feedback from all the reviewers on MV2Cyl. The thorough and insightful comments have significantly contributed to the improvement of our work.

We have carefully considered each question and suggestion and have provided detailed responses to the comments individually. We have compiled the qualitative results discussed in our rebuttal in the attached PDF file.

---

### Decision · Program_Chairs · 2024-09-25

**Decision:**

Accept (poster)

**Comment:**

After the rebuttal and extensive discussion, the reviewers' consensus was to accept the paper given its strengths i.e., the novelty of the proposed extrusion-based 3D reconstruction method using primitives and strong results. The AC recommends acceptance. The AC strongly encourages the authors to incorporate material from the rebuttal and discussion to improve the final version of the paper.